# Leaf-venation-directed cellular alignment for macroscale cardiac constructs with tissue-like functionalities

Mao Mao[1,2,3], Xiaoli Qu[1,2,3], Yabo Zhang[1,2,3], Bingsong Gu[1,2], Chen Li[1,2], Rongzhi Liu[1,2], Xiao Li[1,2], Hui Zhu[1,2], Jiankang He[1,2] ✉ & Dichen Li[1,2]

Recapitulating the complex structural, mechanical, and electrophysiological properties of native myocardium is crucial to engineering functional cardiac tissues. Here, we report a leaf-venation-directed strategy that enables the compaction and remodeling of cell-hydrogel hybrids into highly aligned and densely packed organizations in predetermined patterns. This strategy contributes to interconnected tubular structures with cell alignment along the hierarchical channels. Compared to randomly-distributed cells, the engineered leaf-venation-directed-cardiac tissues from neonatal rat cardiomyocytes manifest advanced maturation and functionality as evidenced by detectable electrophysiological activity, macroscopically synchronous contractions, and upregulated maturation genes. As a demonstration, human induced pluripotent stem cell-derived leaf-venation-directed-cardiac tissues are engineered with evident structural and functional improvement over time. With the elastic scaffolds, leaf-venation-directed tissues are assembled into 3D centimeter-scale cardiac constructs with programmed mechanical properties, which can be delivered through tubing without affecting cell viability. The present strategy may generate cardiac constructs with multifaceted functionalities to meet clinical demands.

Successfully engineering functional cardiac tissues holds great promise in disease modeling, cardiotoxicity testing, and regenerating the damaged myocardium of millions of patients[1–4]. As a complex and efficient pump, the mammalian heart possesses some pivotal characteristics in its structures and function: (i) The highly aligned and densely packed cardiomyocytes (CMs) induce intercellular connections that coordinate electromechanical activity and contractile force of the myocardium[5,6]; (ii) The hierarchical vascular networks transport the extensive oxygen and nutrients efficiently to meet the extremely high-metabolic demands of the myocardium[7,8]; (iii) Biomimetic anisotropic mechanical properties of the myocardium with different effective stiffnesses in the circumferential and longitudinal directions to support the heart's diastole and systole[9,10]. Therefore, ideal cardiac constructs should resemble those crucial structural, physiological, and functional properties of native cardiac tissues[11]. Besides, there are other requirements for the potential therapeutic application, such as clinically relevant size, extracellular matrix (ECM) microenvironment, and proper strength allowing for manipulation and in vivo implantation[12].

Considering the high correlation between structural organization and biological function of the native myocardium, extensive efforts have been invested over the past decades in reproducing cardiac constructs with biomimetic extracellular architecture or multiple components for eliciting myocardial tissue-like functionality[13,14]. Previously, numerous synthetic scaffolds consisting of highly aligned microfabricated grooves or electrospun micro/nano-fibers have been

[1]State Key Laboratory for Manufacturing Systems Engineering, Xi'an Jiaotong University, Xi'an 710049, PR China. [2]NMPA Key Laboratory for Research and Evaluation of Additive Manufacturing Medical Devices, Xi'an Jiaotong University, Xi'an 710049, PR China. [3]These authors contributed equally: Mao Mao, Xiaoli Qu, Yabo Zhang. ✉e-mail: jiankanghe@mail.xjtu.edu.cn

developed to mimic the anisotropic myocardial architecture, providing the topographical cues to direct cell orientation[15-18]. However, those scaffolds lack the ECM biological cues critical for cellular integration and fail to rebuild the biomimetic anisotropic stiffness and hierarchical vasculature. Furthermore, cell alignment induced by anisotropic fibrous scaffolds alone has a limited effect on improving cardiomyocyte maturation[19]. Cell sheet or hydrogel molding approaches, such as seeding mixtures of cell/ECM hydrogel into nylon frames, enabled the generation of large-scale human cardiac constructs from human pluripotent stem cell-derived cardiomyocytes (hiPSC-CMs) with densely packed population and tight cell–cell junctions[20,21]. The self-assembly of cell/matrix mixtures under the guidance of micro-patterned micropillars further allowed for the generation of aligned cellular architectures in simple tissue models[22,23], but such hydrogel-based tissues failed to recapitulate the biomimetic mechanical and cellular heterogeneity (e.g., vascular components) in a controllable manner. Recent advances in the bioprinting of cell/hydrogel bioinks have provided opportunities to fabricate cellular constructs with CMs and embedded vascular structures as predesigned architectures[24-26]. Still, printing tissue constructs with the in vivo-like highly aligned and densely packed cellular arrangement, microscale hierarchical vascular structure, and sufficient mechanical property remains challenging. In addition, the cell/hydrogel-compaction-induced force may induce a dramatic change in the predefined structure of the printed CM constructs. In conclusion, these signs of progress in cardiac tissue engineering have improved the ability to recapitulate the more nuanced characteristics of the native myocardium from different aspects. However, the combined capabilities to generate a large-scale functional cardiac tissue with controlled cell orientation, mechanical anisotropy, and hierarchical pre-vascular networks in a single strategy have not yet been achieved.

To address the aforementioned challenges, we report a leaf-venation-directed (LVD) strategy for generating pre-vascularized, functional cardiac constructs that enable recapitulating the anisotropic structural and electrophysiological characteristics of the native myocardium. Leaf venation, a nature-optimized hierarchical network consisting of close-packed cells and aligned fibers, can work as a critical highway for the transmission of action potentials, which resembles the electrophysiological activity in the mammalian myocardium. Besides, leaf venation is remarkably similar to mammalian vasculature regarding the hierarchical structure and high-efficient fluid transportation function[27]. Inspired by these similarities, we employed the leaf-venation networks to direct the self-assembly of cell/hydrogel hybrids into tissue constructs with a physiologically relevant close-packed arrangement and robust functionalities. It was demonstrated that the LVD highly aligned and densely packed arrangement benefited the differentiation, maturation, beating, and electrophysiological activities of the engineered cardiac tissues, evidenced by detectable electrophysiological activity, regular and synchronous calcium ion ($Ca^{2+}$) puffs, noticeable local field potential, and upregulation of CM-specific proteins and maturation genes. Furthermore, with the support of the electrohydrodynamic (EHD)-printed scaffold, the engineered LVD tissues were transferred and assembled into 3D pre-vascularized cardiac constructs with programmed mechanical properties. The 3D constructs with a clinically relevant size could be rolled up and delivered through a tube and then recover their initial shape following injection without affecting cellular viability and structural integrity, demonstrating their potential for minimally invasive implantation.

## Results

### The strategy for engineering LVD tissues using HUVECs
The critical point to generate functional cardiac and vascular tissues is to recapitulate the physiologically relevant, highly aligned and densely packed cellular arrangement, eliciting the tight intercellular connections that govern their concerted biological activity[1,28]. To achieve this,

we propose an LVD strategy using the microchannels derived from leaf-venation networks as geometric confinement to guide the morphological evolution of high-density cells in ECM hydrogel. As described in our previous works, polydimethylsiloxane (PDMS) substrates with leaf-venation-inspired microchannels, consisting of a primary channel and multiple branch channels, can be produced from the skeleton of leaf-venation networks (Fig. 1a(i))[29]. The amphiphilic treatment is conducted to prevent the adhesion between PDMS and cell/hydrogel matrices (Fig. 1(a(ii)). Cell-laden fibrin hydrogel precursor solution is then added and gelled within the microchannels of the PDMS substrate (Fig. 1a(iii)). The uniformly encapsulated cells begin to spread and exert traction forces on the surrounding matrix[30], initiating LVD morphological evolution process (Fig. 1a(iv,v)). Subsequently, the cell-spreading-induced forces can induce the shrinkage of the cell/hydrogel[31], which detaches from the amphiphilic-treated PDMS surfaces and self-organize into densely packed tissues (Fig. 1a(vi)). This process eventually leads to interconnected tissue bundles aligned along the microchannels (Fig. 1a(vii)). As a control, the cells cultured in non-amphiphilic-treated PDMS substrate spread in the matrix but could not detach from the microchannels, leading to their random distribution (Supplementary Fig. 1). As a demonstration, we successfully employed the LVD strategy to engineer a large-scale pre-vascular tissue, with interconnected hierarchical networks confined in the leaf-venation-inspired microchannels (Fig. 1b). The overall size of the generated oval leaf-like LVD tissue is ~760 mm², with a length of ~45 mm, a maximum width of 24 mm, and the cellular bundles are ~20-50 μm in diameter. It was found that the mixture of human umbilical vein endothelial cells (HUVECs) and fibrin hydrogel precursor solution detached from the channels immediately postseeding, and gradually compacted in the microchannels. The randomly distributed round cells eventually became a stable, interconnected cellular bundle with smooth borders after 48 h of culture (Fig. 1c). On the contrary, HUVEC/fibrin hydrogel cultured in the control group remained random and anchored to the channel surface, since the cell-spreading-induced force was not enough to break down the adhesion between the non-amphiphilic-treated PDMS channels and cell-laden fibrin hydrogel (Fig. 1d). Scanning electron microscopic (SEM) images also confirmed that the highly elongated and compacted morphology of cell bundles along the microchannels in LVD tissues (Fig. 1e). Due to significant hydrogel compaction and tissue remodeling, the average width of the HUVEC/fibrin hydrogel was reduced ~2-fold in the primary channel and ~5-fold in branch channel (Supplementary Fig. 2).

Immunofluorescence staining was performed to characterize the cellular arrangement and functional protein expression within LVD-HUVEC tissues. Cytoskeletal staining demonstrated that HUVECs in LVD tissues reached the confluence with a highly elongated morphology (Fig. 1f). The quantitative results revealed that most cells were highly aligned along the longitudinal direction of the microchannels (Fig. 1h). In contrast, the cells in control groups were randomly oriented in an irregular morphology (Fig. 1g, h). A high level of CD31, a specific endothelial marker of platelet-endothelial cell adhesion molecule, was expressed in regions of cell–cell contact throughout the entire LVD-HUVEC tissues, with no noticeable differences in the varying channel regions. This demonstrated that the densely packed LVD tissues benefited the formation of tight intercellular junctions between neighboring cells and the maintenance of the functional endothelial phenotype (Fig. 1i). The intercellular junctions and pinocytotic vesicles found in the ultrastructural analysis further indicated the normal endothelial function in transporting biological molecules (Fig. 1k). In contrast, the cells in the control groups expressed a lower level of CD31 in an irregular pattern (Fig. 1j). Another essential characteristic of LVD tissues is their interconnected tubular structures in all the primary and branch channels and the bifurcation (Fig. 1l). At the same time, the cells in control were distributed at the bottom of microchannels (Fig. 1m). These self-

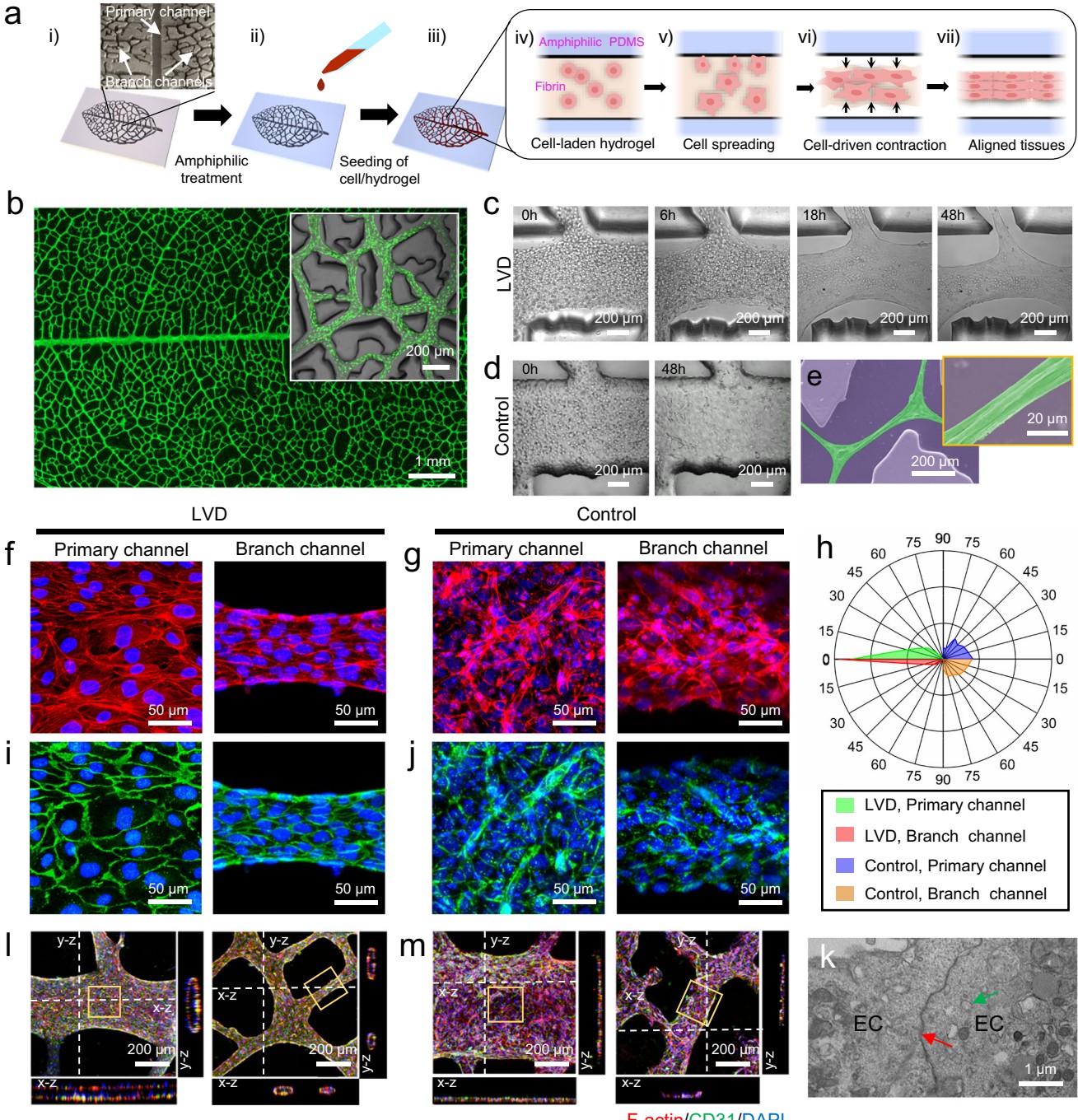

**Fig. 1 | LVD strategy of generating large-scale tissues with highly aligned populations and tight intercellular connections using HUVECs. a** Schematic illustration of the seeding and self-assembly of cells/hydrogel into highly aligned and densely packed cellular bundles directed by leaf-venation-inspired microchannels. **b** Stitched fluorescent image shows the viability of a large-scale LVD tissue consisting of hierarchical cellular bundles confined in leaf-venation-inspired microchannels (inset). **c** The time course of self-assembly of cells/hydrogel into aligned cellular bundles with smooth borders during 48 h of culture within primary channels. **d** No noticeable morphological change of cell-laden fibrin hydrogel in the non-amphiphilic-treated PDMS channels as control. **e** Representative SEM images of LVD cellular bundles (green) within branch channels (purple) cultured for 3 days. The inset is a magnified image of a cellular bundle. The stained F-actin in LVD (**f**) and control tissues (**g**), and corresponding histogram showing the quantitative nuclear alignment relative to the preferred nuclear orientation (**h**). The stained biomarker CD31 in LVD (**i**) and control tissues (**j**). **k** Transmission electron micrograph showing intercellular junctions (red arrow) and pinocytotic vesicles (green arrow). Confocal files showing the tubular structures of LVD (**l**) and control (**m**) tissues. **f**, **g**, **i**, and **j** zoomed-in views of the framed regions in (**l**) and (**m**), respectively.

assembled tubular structures were in accordance with previous studies, which produced endothelial tubes with simple geometry and proved their value as templates to trigger the formation of new capillaries in vivo in the prescribed pattern[32,33]. Our LVD-HUVEC tissues possess much more hierarchical networks, which might benefit the formation of biomimetic vasculatures in vivo.

## The dynamic formation process of LVD-HUVEC tissues
We next sought to characterize the dynamic self-assembly process of LVD-HUVEC tissues from uniformly distributed cells/hydrogel into the highly aligned and densely packed tubular structures (Fig. 2a). The cells and fibrin hydrogel composites cultured at different time points were visualized with a laser scanning confocal microscope (Fig. 2b).

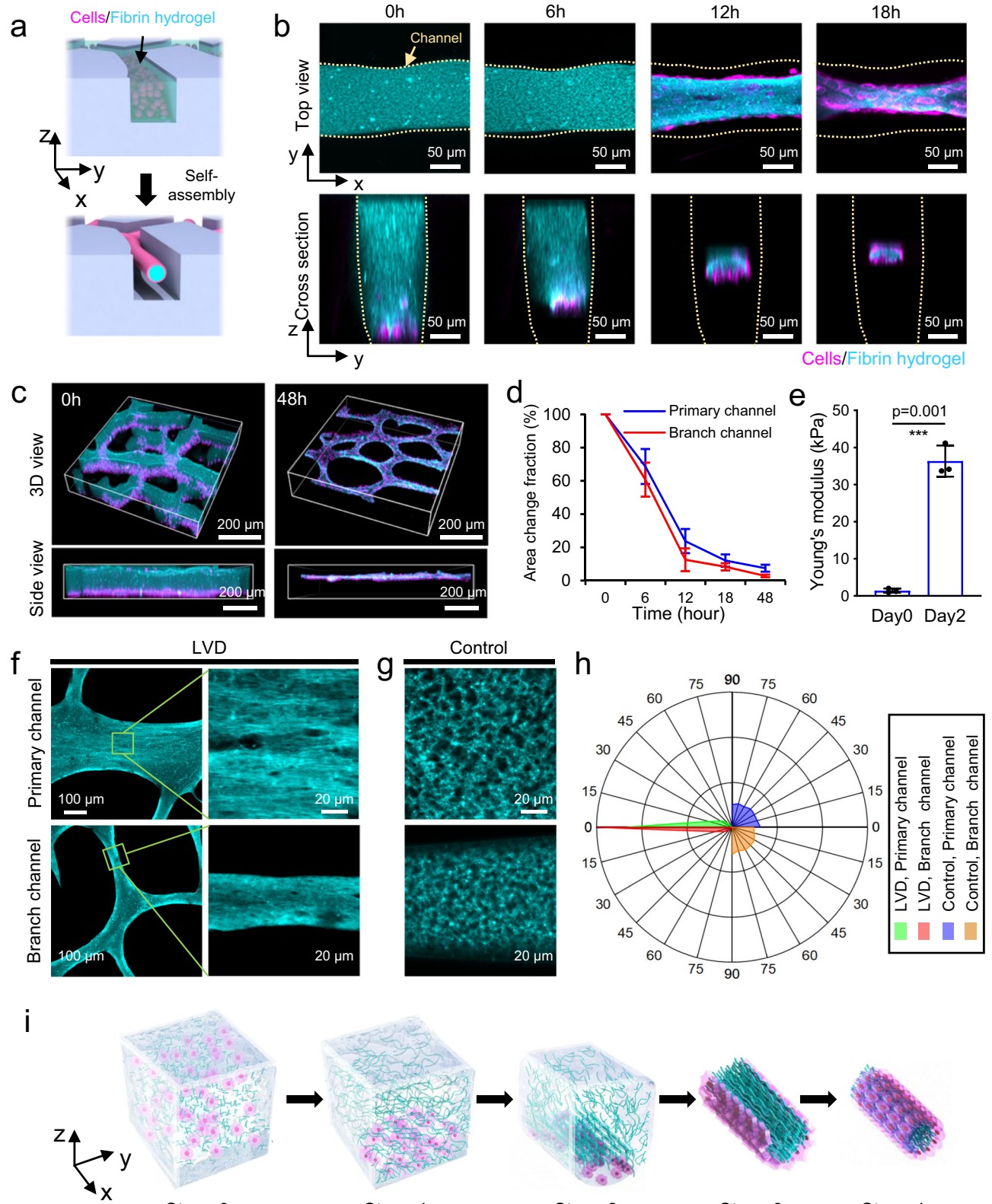

**Fig. 2 | Investigation of the formation process of LVD-HUVEC tissues featured with dense package and tubular structures. a** Schematic illustration of cells/hydrogel self-assembly into tubular structure directed by microchannels. **b** The sequential steps of LVD self-assembly of HUVECs (magenta)/fibrin hydrogel (cyan) in branch channel. **c** The randomly distributed cells/fibrin transformed into continuous and interconnected tubular structures. **d** The change of the cross-sectional area fraction of the cells/fibrin hydrogel relative to the channel. $n = 4$ independent samples. **e** Average Young's modulus of LVD tissues postseeding and at day 2. $n = 3$ independent samples. The microstructure of fibrin hydrogel fibers in LVD (**f**) and control (**g**) tissues, and corresponding histogram showing the quantitative fiber alignment relative to the preferred fiber orientation (**h**). **i** Schematic illustration summarizing five stages of LVD tissue formation. Data are represented as mean values ± standard deviation. The significant difference is determined by unpaired two-tailed $t$-test, 95% confidence interval. ***$p < 0.001$.

After seeding into the microchannel, all the HUVECs were settled in the bottom of the crosslinked fibrin hydrogel. At the beginning 6 h of culture, a decrease in the thickness of fibrin hydrogel was found, indicating that the cells had started to spread and exerted contractile forces on the surrounding ECM hydrogel. In the next 6 h, the fibrin hydrogel experienced a sudden massive shrinkage in the cross-section, especially in the thickness direction, and was compacted into a thin rod-like structure. While most cells were still in the bottom of the hydrogel, some had begun to line along the out surface of the condensed fibrin rod. After 18 h, the cells migrated from the bottom onto the top surface, forming a confluent, tubular endothelial structure filled with densely packed fibrin. After 48 h, the tubular structure was further compacted into a smaller dimension (Supplementary Fig. 3). Eventually, in all the microchannels, including the primary and branch channels, the randomly distributed cells/fibrin transformed into continuous and interconnected tubular structures with densely packed fibrin encircled by a cellular layer (Fig. 2c, Supplementary Fig. 4). Indeed, the interconnected LVD cellular bundles were suspended between the microchannels rather than settling at the bottom. In contrast, no apparent morphological change of the cells and fibrin hydrogel was found in the control group after 48 h of culture (Supplementary Fig. 5). The quantitative results indicated that the average cross-sectional area fraction of the cells/fibrin hydrogel was reduced ~13-fold in the primary channel and ~33-fold in the branch channels, during the progression of LVD tissue development (Fig. 2d). This significant decrease in volume is expected to induce the improved mechanical property of the cells/fibrin hydrogel composites. Therefore, atomic force microscopy (AFM) nanoindentation was conducted to measure the local stiffness of the cells/fibrin hydrogel mixture and the final resultant LVD-HUVEC tissues[34]. After 48 h of culture, the mean local Young's modulus of the constructs was increased ~45-fold, from roughly 0.8 kPa to ~37.3 kPa (Fig. 2e). Meanwhile, the densely packed fibrin fibers in LVD tissue were oriented along the microchannels (Fig. 2f), while that in the control group remained in porous structures and randomly distributed (Fig. 2g) after 48 h of culture. The quantitative results demonstrated that the fibrin fiber alignment in the LVD and control tissues (Fig. 2h) followed the cellular alignment pattern along the longitudinal channel directions (Fig. 1h). The LVD-HUVEC tissues were then sandwiched by gelatin gel and perfused with nanobeads solution. The result demonstrated the occlusion of the tubular structures, which should be owing to their heavily compressed fibrin fibers inside (Supplementary Result).

These data reveal a self-organization process of LVD tissues through six consecutive stages (Fig. 2i): Stage 0, cells and fibrin precursor solution fill the microchannels (oriented along the y-axis) uniformly. Stage 1, the round cells settle in the bottom of the crosslinked fibrin hydrogel, consisting of randomly distributed fibers. Stage 2, cells spread and exert traction force on the surrounding ECM and cells, which induces them to gather towards the central region gradually; during this process, a strain field is formed within the hydrogel along the longitudinal direction of the microchannel (y-axis), which guides the local alignment of cells and fibrin fibers in the same direction. Stage 3, Under the combined action of the cellular force and the longitudinal strain force, the fibrin fibers are compressed into a highly aligned rod-like structure; cells gradually migrate from the bottom to cover the surface of the compressed fibrin fibers. Stage 4, The fibrin fibers are further compressed; the cells reach confluence encircling the dense fibrin fibers and orient along the longitudinal direction in a highly aligned arrangement.

## Structural maturation and molecular maturation of LVD-neonatal-rat-cardiac tissues

We further investigate the potential of the LVD strategy to engineer highly aligned and densely packed cardiac tissues and the contribution of this biomimetic tissue anisotropy to myocardial tissue-like maturation and functionality. After 5 days of culture, the seeded mixture of neonatal rat CMs and fibrin hydrogel self-assembled into the tissue constructs defined by the leaf-venation pattern (Fig. 3a). Higher magnification of the LVD tissues revealed that the CMs were densely and uniformly packed, and strongly elongated within the interconnected tissue bundles in all the channels (Fig. 3b); the control tissue showed random cellular orientation (Fig. 3c). The cross-section of the confocal file indicated that the CMs also compacted to form a tubular structure (Fig. 3d). The local stiffness of the 5-day-old LVD-cardiac tissues, measured by AFM nanoindentation, was ~40 kPa, which was close to that of the rat adult myocardium (~22-51 kPa)[35]. This biomimicking stiffness might be an essential contributor to the differentiation, hypertrophy, and electromechanical function of the CMs in the aligned LVD-cardiac tissues[36,37].

The maturation of LVD-cardiac tissues was assessed using immunostaining staining of CM-specific markers, including contractile protein sarcomeric α-actinin and the gap junctional protein connexin 43 (CX43). The 5-day-old LVD-cardiac tissue construct exhibited highly aligned and interconnected striated sarcomeric structures, as well as higher expression and uniform distribution of CX43 compared to the control tissues with the random and loose cellular distribution (Fig. 3e, f). The quantified data based on the immunofluorescence staining indicated a longer sarcomere length and a higher level of CX43 in LVD-cardiac tissue (no evident difference between the primary and branch channels) than that in the control tissues (Fig. 3g, h). In addition, the expression of maturation-related cardiac marker gene was investigated. After 5 days of culture, higher expression of electrical coupling-related (KCNJ2, GJA1)[38,39] and metabolic-related genes (PDK4, CPT1B, PPARGCLA)[40,41] was found compared to the control tissue (Fig. 3i), which results were consistent with the morphometric analysis (Fig. 3e–h). All these results demonstrated that the LVD-cardiac tissues with highly aligned and densely packed structures benefited CM maturation with improved cell–cell coupling and contractile phenotype.

## Electrophysiological functions of LVD-neonatal-rat-cardiac tissues

Calcium transients related to excitation-contraction coupling were investigated to evaluate the synchronous contraction functions of the LVD-rat-cardiac tissues. Within each recorded tissue area, the fluorescent intensity of calcium spikes for five locations of interest was plotted over 8 s. The resultant calcium transients in both the primary and branch channels displayed strong and synchronous calcium mobilization, indicating the synchronous beating of the LVD tissues, which was visible under the naked eye on day 5 of culture (Fig. 4a, c, Supplementary Movies 1 and 2). On the other hand, disorderly and asynchronous $Ca^{2+}$ puffs without rhythmic patterns took place in the control tissues (Fig. 4b, d, Supplementary Movies 3 and 4). Furthermore, the quantitative parameters based on calcium transients demonstrated less time to reach the calcium peak and higher beating frequency in the LVD tissue (no evident difference between the primary and branch channels) compared with the control tissue (Fig. 4e,f).

The LVD-rat-cardiac tissues with highly aligned and densely packed structures enabled the production of high-amplitude extracellular field potentials. To demonstrate it, we integrated two thin platinum wires as conductive microelectrodes into the primary channel of LVD system for in situ biologically relevant electrophysiological monitoring of CMs (Fig. 4g(i)). During the formation of LVD-cardiac tissues, the microelectrodes could be enclosed by the densely packed cellular bundle (Fig. 4g(ii)). Through this system, we detected the extracellular field potentials of 2-day-old LVD-rat-cardiac tissues in ~40 mV spaced with a frequency of ~1 Hz, whose amplitude increased to ~60 mV with an improved frequency of ~2-3 Hz after 4 days of culture (Fig. 4h, i). In contrast, no electrical signal could be detected in the control tissue. This implies that our LVD system with built-in

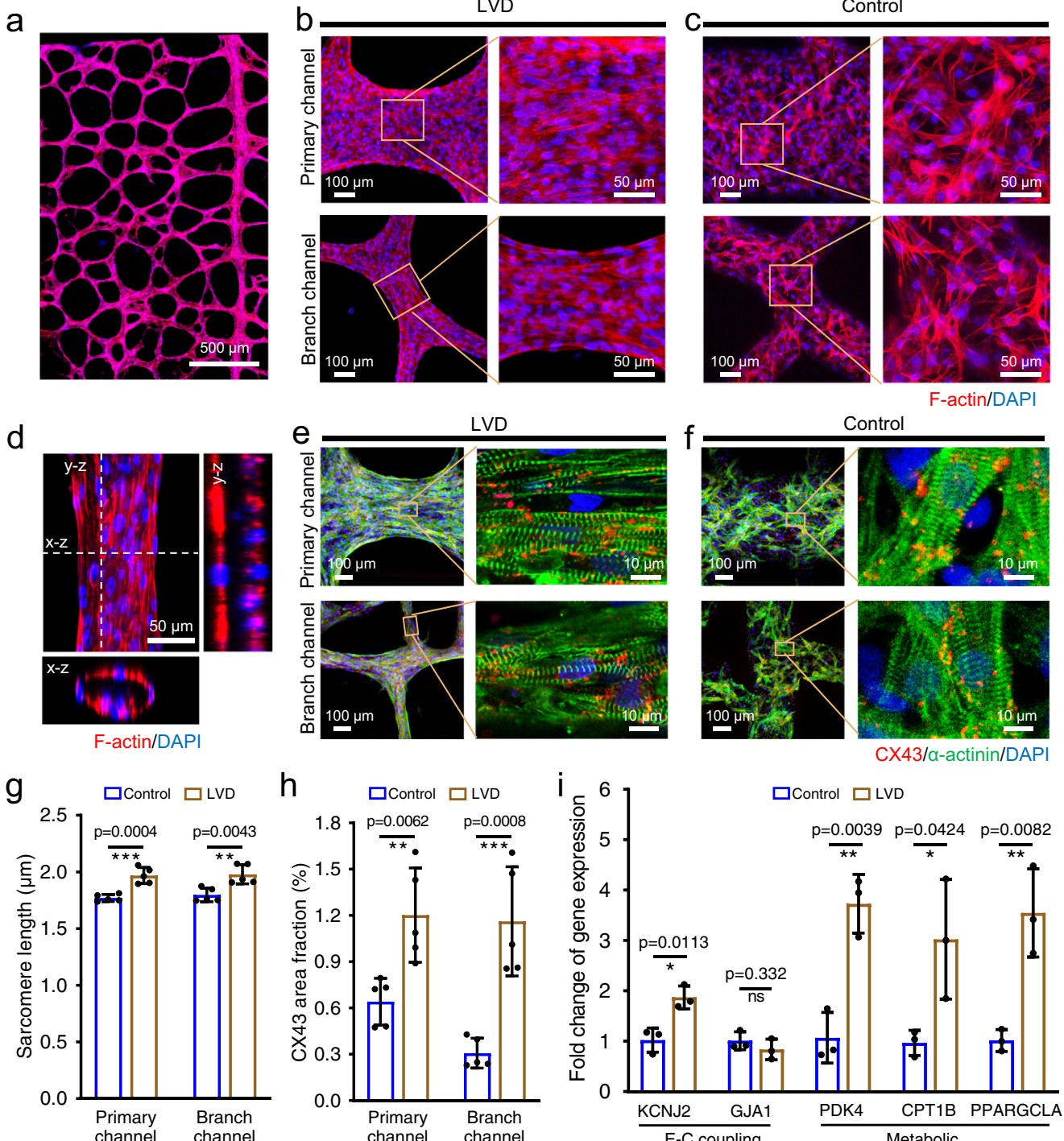

**Fig. 3 | Structural characterization and maturation of the engineered LVD-rat-cardiac tissues.** F-actin staining of 5-day-old LVD-cardiac tissue (**a**), and the representative images in different locations of LVD (**b**) and control tissues (**c**). **d** Confocal images of LVD-cardiac bundles showing the tubular structures. Representative images of rat-CM proteins expression stained for sarcomeric α-actinin (green), CX43 (red), and DAPI (blue) in LVD (**e**) and control (**f**) tissues after 5 days of culture. Quantitative analysis of sarcomeric length (**g**) and CX43 area fraction (**h**). $n = 5$ independent samples. **i** Relative gene expression in LVD tissues vs. control tissues. $n = 3$ independent samples. Data are represented as mean values ± standard deviation. The significant difference is determined by unpaired two-tailed $t$-test, 95% confidence interval. *$p < 0.05$, **$p < 0.01$, ***$p < 0.001$ and ns ($p > 0.05$) no significant difference.

microelectrodes enables noninvasive monitoring of the cardiac tissues' natural maturation and other rhythmic cellular phenomena, showing the potential in online cardiotoxicity testing or functional cardiac tissue engineering. Then, the LVD system's ability to monitor changes in the frequency of electrical signals in response to the supplement of the drug was investigated. For example, the field potential recording revealed that the signal frequency reached 1.78 Hz from an initial value of 1.13 Hz without a significant change or degradation in signal quality following adding 5 μM isoproterenol to the culture (Fig. 4j). Besides, electrical stimulation could also be applied through the two built-in microelectrodes to pace the CMs. The interfering by applying acute electrical stimulation (10 V, 50 ms) at different frequencies (1 and 2 Hz) activated the cells throughout the LVD-cardiac tissues and modulated their beating activities (Fig. 4k, l).

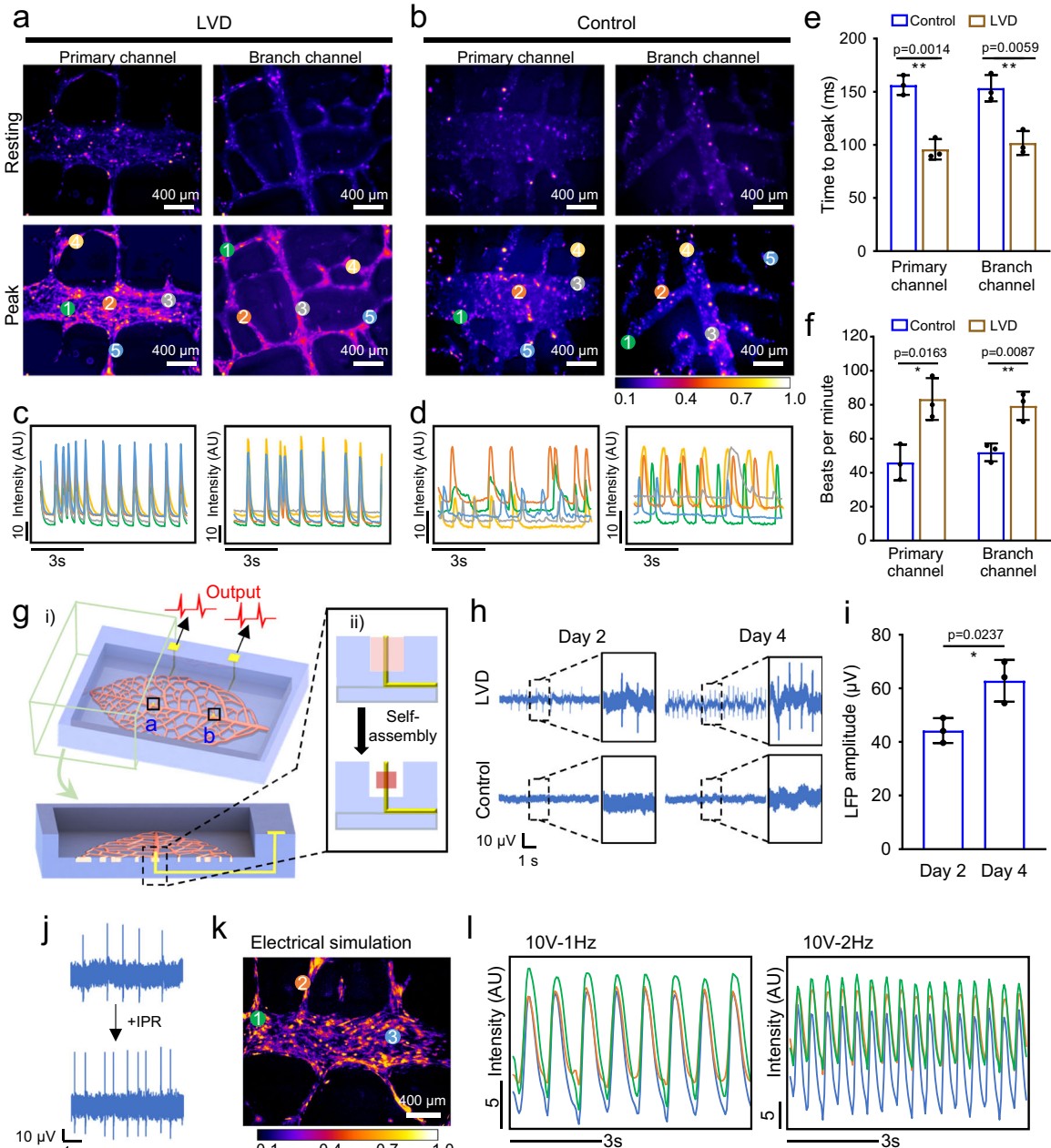

**Fig. 4 | Electrophysiological functions of the engineered LVD-rat-cardiac tissues.** Fluorescence micrographs of calcium transients showing the regions of interest in LVD (**a**) and control (**b**) tissues. Quantification of calcium transients in LVD (**c**) and control (**d**) after 5 days of culture (separate regions of interest are represented in different colors). AU indicates arbitrary units. **e** The quantification of time to peak of calcium transient correspondingly. $n = 3$ independent samples. **f** The quantification of beating rate in spontaneously beating LVD and control tissues. $n = 3$ independent samples. **g** Schematic of customized in situ electrical recording/stimulation system. **h** Representative field potentials recorded from the 2-day-old LVD tissue by the two integrated microelectrodes with the distance gap of 10 mm. **i** Electrophysiological data collected in the LVD (top row) and control (bottom row) tissues. $n = 3$ independent samples. **j** Local field potential change before and after the addition of isoproterenol (5 µM). **k** Fluorescence micrograph of calcium transients showing the regions of interest in LVD tissue under electrical stimulation. **l** Quantification of calcium transients (as quantified through normalized fluorescence intensity) with a 10 V, 50 ms, 1 Hz, and 2 Hz pacing regime (separate regions of interest are represented in different colors). Data are represented as mean values ± standard deviation. The significant difference is determined by unpaired two-tailed $t$-test, 95% confidence interval. *$p < 0.05$, **$p < 0.01$.

Therefore, these results indicate that the present LVD strategy enables engineering functional cardiac tissues with interconnected, highly aligned, and densely packed structures, thus benefiting their in vivo-like cellular phenotype, synchronous beating activities, and electrophysiological functions. In addition to a therapeutic approach, the LVD system can be exploited for in vitro studies, such as drug screening assays in a 3D microenvironment, as well as in situ monitoring of electrophysiological activities of the cardiac tissues.

## Engineering functional LVD-human-cardiac tissue from hiPSC-CMs

While the above results have comprehensively shown the advancement of the LVD strategy for producing rat-cardiac tissues with hallmarks of the native myocardium, neonatal rat CMs cannot be utilized for clinical therapeutics. Engineering human cardiac tissues based on hiPSC-CMs, avoiding issues of controversial ethical value and limited expansion and regeneration capacity of

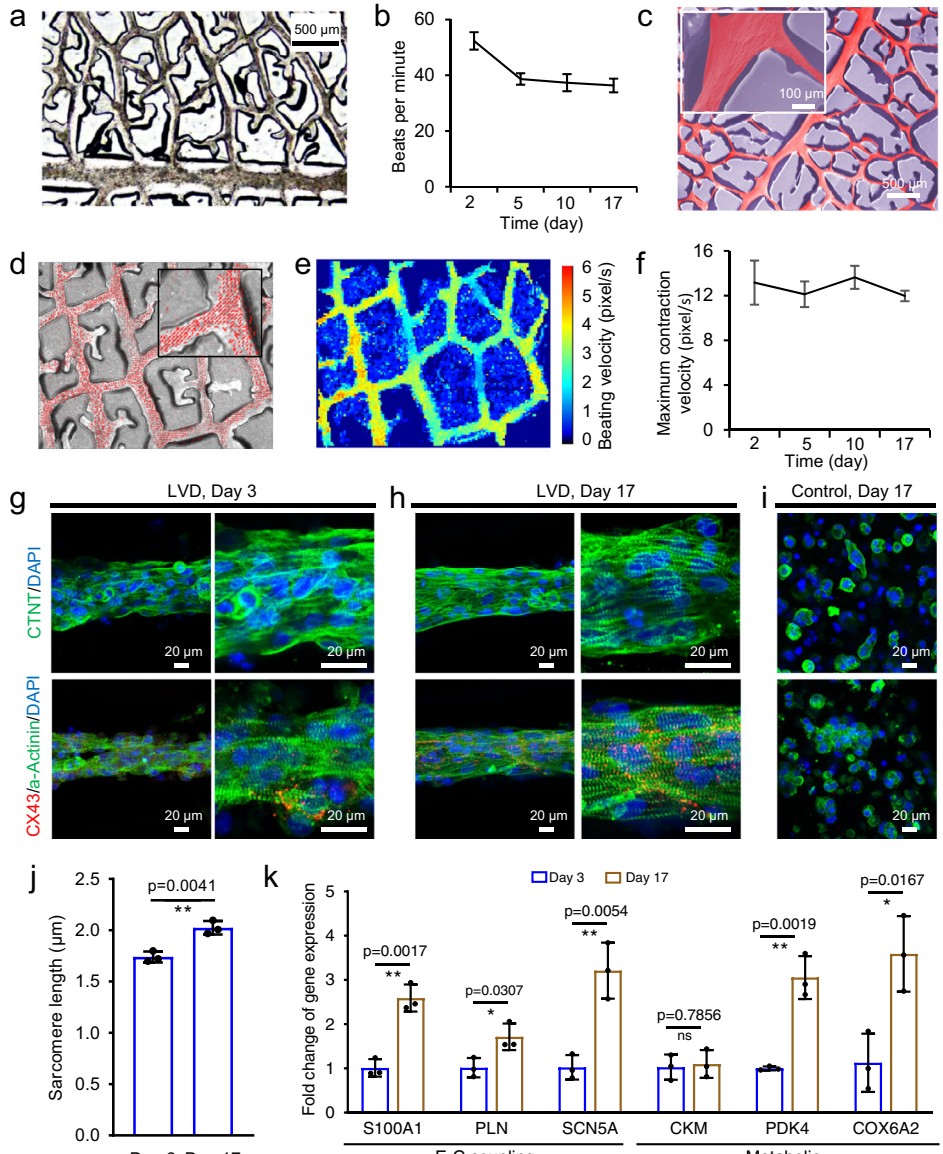

**Fig. 5 | Characterization of the engineered LVD-human-cardiac tissues from hiPSC-CMs. a** Bright-field image of LVD tissue in microchannels after 2 days of culture. **b** Quantification of beating rate of LVD tissue at different time points. *n* = 3 independent samples. **c** SEM image of LVD tissue in microchannel after 17 days of culture. Inset: a higher-magnification image of LVD bundle. **d**–**f** Contractile motion tracking of beating LVD tissues in microchannels. **d** Representative frame overlayed with the respective motion vectors. Inset: higher-magnification image showing the motion vectors of LVD tissue. **e** Heat map of localized mean contraction velocities. **f** Maximum contraction velocities of beating LVD tissues at different time points. *n* = 3 independent samples. Sarcomeric structures/CX43 and CTNT staining of cellular bundles (left row) and their higher-magnification view (right row) in LVD tissues after 3 days (**g**) and 17 days (**h**) of culture, compared with that in control tissues after 17 days (**i**) of culture. **j** Quantitative analysis of sarcomeric length in LVD tissues. n = 3 independent samples. **k** Relative gene expression in 3-day-old vs. 17-day-old LVD-human cardiac tissues. Data are represented as mean values ± standard deviation. The significant difference is determined by unpaired two-tailed *t*-test, 95% confidence interval. **p* < 0.05, ***p* < 0.01, and ns (*p* > 0.05) no significant difference.

primary CMs, have been endowed with great expectations to provide solutions[3]. One crucial concern is to drive their structural and functional maturation, evidenced by elongated morphology, stable contractile machinery, gene expression profiles, and electrophysiology[2,3].

Therefore, we continued investigating the generation of matured human cardiac tissues by culturing hiPSC-CMs using the LVD strategy over a 17-day culture. A small amount of individually beating CMs was observed within 24 h after seeding. During the following 2 days, hiPSC-CMs/fibrin hydrogel remodeled and decreased in width, resulting in a large-scale LVD-hiPSC-derived-cardiac tissue with interconnected cellular bundles directed by the microchannels (Fig. 5a). Meanwhile, the initial asynchronous beating converted to a coherent, synchronous

macroscopically visible contractions (Supplementary Movie 5). From day 2 to day 5, these synchronous contractions gradually decreased from 50–60 to 35–40 beats per minute and then stayed at a consistent and stable beating rate in the following days, indicating long-term electrophysiological coupling of the CMs (Fig. 5b). After 17 days of culture, the LVD-human-cardiac tissue remained synchronous macroscopically visible contractions and relatively stable configuration with interconnected bundles confined in the microchannels (Fig. 5c). The video-based analysis of contractile motion in LVD tissues demonstrated their longitudinal contraction direction along the microchannels (Fig. 5d), relatively uniform contraction velocity between different regions (Fig. 5e), and stable contraction velocities over 17 days of culture (Fig. 5f).

Immunofluorescent staining of the cardiac tissues was performed to assess hiPSC-CM microscale organization of myofibrils and intercellular coupling. The initially formed LVD-cardiac tissues after 3 days of culture had shown a relatively densely packed arrangement with significant tendencies in alignment, while some cells are still close to round morphology (Fig. 5g). After 17 days of culture, LVD-cardiac tissues demonstrated a more regular sarcomeric α-actinin and cardiac troponin T (cTnT) organization with clear contours. Nearly all the cells were elongated in a high aspect ratio, resulting in highly aligned and densely packed tissue structures (Fig. 5h). The cross-section of the confocal file indicated that the hiPSC-CMs also form a tubular structure (Supplementary Fig. 6). In contrast, the 17-day-old hiPSC-CMs cultured in non-amphiphilic-treated PDMS remained in round morphology (Fig. 5i). The sarcomere length of the 17-day-old CMs in LVD-cardiac tissues displayed a statistically significant increase compared with that of day 3, extended from ~1.74 μm to ~2.02 μm, whose value was quite close to that in adult human CMs (2.2 μm)[42] (Fig. 5j). In addition, the 17-day-old LVD-cardiac tissues expressed a higher level of CX43. These results indicated that more extended periods of culture could benefit the structural maturation of LVD-cardiac tissues.

In addition to the structural proteins of sarcomere and cTnT, we further sought to reveal the molecular signatures underlying the advanced functional maturation of LVD tissues. 6 cardiac marker genes were chosen, including two groups: excitation-contraction coupling (*S100A1*, *PLN*, *SCN5A*) and metabolic (*CKM*, *PDK4*, *COX6A2*)[20,43,44], thus reflecting key maturation processes in developing hiPSC-derived CMs. 5 of the 6 genes progressively increased at day 17 in LVD tissues compared to day 2, suggesting enhanced LVD-hiPSC-derived-cardiac tissue maturation over time in culture (Fig. 5k).

## Engineering of 3D pre-vascularized cardiac tissues with programmed mechanical property

Engineering 3D functional cardiac tissues with a pre-vasculature and proper mechanical property are pivotal to promoting their long-term survival and serving as temporary mechanical support to prevent the progression of postinfarction left ventricular remodeling, thus benefiting the restoration of the heart's normal contraction behavior and function in vivo[9,45]. For this purpose, we develop a scaffold-assisted method to achieve the transfer of LVD-cardiac and LVD-HUVEC tissues and assembly of them into 3D tissue composites with the programmed mechanical property. The elastic scaffold with precisely defined serpentine microarchitecture was applied in the follow-up study, which we hypothesized was able to reproduce the anisotropic and viscoelastic behavior of the native myocardium.

Figure 6a shows the transfer and assembly process of the multiple LVD tissues. Specifically, after the removal of the culture medium, an EHD-printed elastic scaffold was placed onto the well-formed LVD-HUVEC tissue in the PDMS substrate. 3 mg/mL bovine fibrin precursor solution was then added and polymerized for 5 min at 37 °C to encapsulate the scaffold and tissue together (Fig. 6a(i,ii)). With the mechanical support of the scaffold, the LVD tissue in fibrin hydrogel could be easily transferred out of the PDMS substrate (Fig. 6a(iii)). This transferred LVD tissue was then placed onto the other well-formed LVD-cardiac tissue, with 100 μL bovine fibrin precursor solution added between them. After polymerizing for 5 min at 37 °C, the fibrin hydrogel could work as glue to bond the LVD tissues together (Fig. 6(a,iv)). The 2-layer LVD tissue could then be obtained (Fig. 6a(v)). By repeating this elastic-scaffold-based transferring and fibrin-glue-based bonding process, the multiple LVD-cardiac and LVD-HUVEC tissues could be assembled into 3D pre-vascularized cardiac tissue constructs (Fig. 6(a,vi)). The finite-element analyses (FEA) calculations were then conducted for the rational structural design of the elastic scaffolds, which could endow 3D pre-vascularized cardiac tissues with directionally dependent mechanical properties along the circumferential (CIRC) and longitudinal (LONG) axe that conforms to the

nonlinear deformation of the native myocardium in plane (Fig. 6(a,vii)). The detailed FEA calculation can be found in Supplementary Fig. 7.

EHD printing was employed to produce the elastic polycaprolactone (PCL) scaffold since it enabled the high-resolution fabrication of microscale serpentine structures comparable to the design, thus granting us freedom in tailoring the mechanical property of assembled tissue constructs[46]. The experimental results demonstrated that the dimensions of the EHD-printed scaffolds agreed with those designed dimensions (Supplementary Fig. 8 and 9). Figure 6b, c, and Supplementary Fig. 10 provide the resultant FEA simulations and macroscopic images of the 3D tissue composite upon uniaxial stretching (from top to bottom: 0%, 10%, and 20%) along the LONG direction, and Supplementary Fig. 11 provides that along the CIRC direction. Figure 6d demonstrates good consistency in stress–strain curves along the CIRC and LONG direction among the corresponding FEA calculations, experimental measurement, and native adult rat heart tissue (the data was acquired from the previous literature[10]), indicating that the combined theoretical calculations and EHD printing can provide an accurate and effective pathway for the generation of the scaffold-reinforced 3D assembled LVD tissues with cardiac-tissue-like nonlinear mechanical behaviors.

Figure 6e shows the transferred monolayer LVD-cardiac tissue from rat CMs after 4 days of culture, demonstrating the structural integrity and stability of the engineered tissue integrated with the serpentine PCL scaffold. As a demonstration, 4-day-old LVD-rat-cardiac and 3-day-old LVD-HUVEC tissues were stained by red and green cell trackers respectively, and then superimposed with each other to generate a 4-layer LVD-cardiac tissue construct with a thickness of ~500 μm (Fig. 6f, g).

## Injectable delivery of 3D LVD-cardiac constructs

To avoid open chest surgery, we further explore the feasibility of minimally invasive implantation of the assembled LVD-cardiac constructs. With the mechanical aid of the EHD-printed elastic scaffolds, the 4-layer cardiac constructs are expected to be injected by a tubing passing through the chest wall and placed onto the diseased epicardium. Before injection, the assembled cardiac constructs were rolled up around a rod with a diameter of 2.5 mm and then transferred into a tubing with an inner diameter of 6 mm (Fig. 7a). The rolled 3D tissues could stay in the tubing stably (Fig. 7b). Under the flowing water, the cardiac constructs could be rushed out of the tubing and recovered to their original shape (Fig. 7c).

As a proof of concept, the assembled cardiac construct was injected onto the ventricle of the ex vivo porcine heart, which shared a similar structure and volume comparable to the human heart. With placement by forceps, the injected cardiac construct was capable of covering a clinically relevant size of the porcine heart with a conformable and tight attachment between them (Fig. 7d). The EHD-printed fiber scaffold and interconnected LVD tissues could preserve their structural integrity after ejection (Fig. 7e). No lamination in the LVD tissues was observed during this whole assembly and injection implantation process, indicating that the assembly strategy based on fibrin hydrogel bonding enables the generation of stable thick tissue structures. Previously, fibrin hydrogel has been used as glue to secure the cardiac patch on the heart in vivo[47]. We then conducted a preliminary study to attach the 3D LVD tissue constructs onto the ex vivo porcine heart using fibrin glue. The result demonstrated firm fixation between them, even when they suffered from rotation and mechanical stretch (Supplementary Movie 6 and Supplementary Fig. 12).

During injection, the applied force is mainly stored in the scaffold, thus facilitating the tissues' return to their original shape; relatively little force in the LVD tissue-laden hydrogel is beneficial for their viability. To investigate this, the FEA calculations were conducted to study the distribution of strain force within the LVD tissue consisting of the

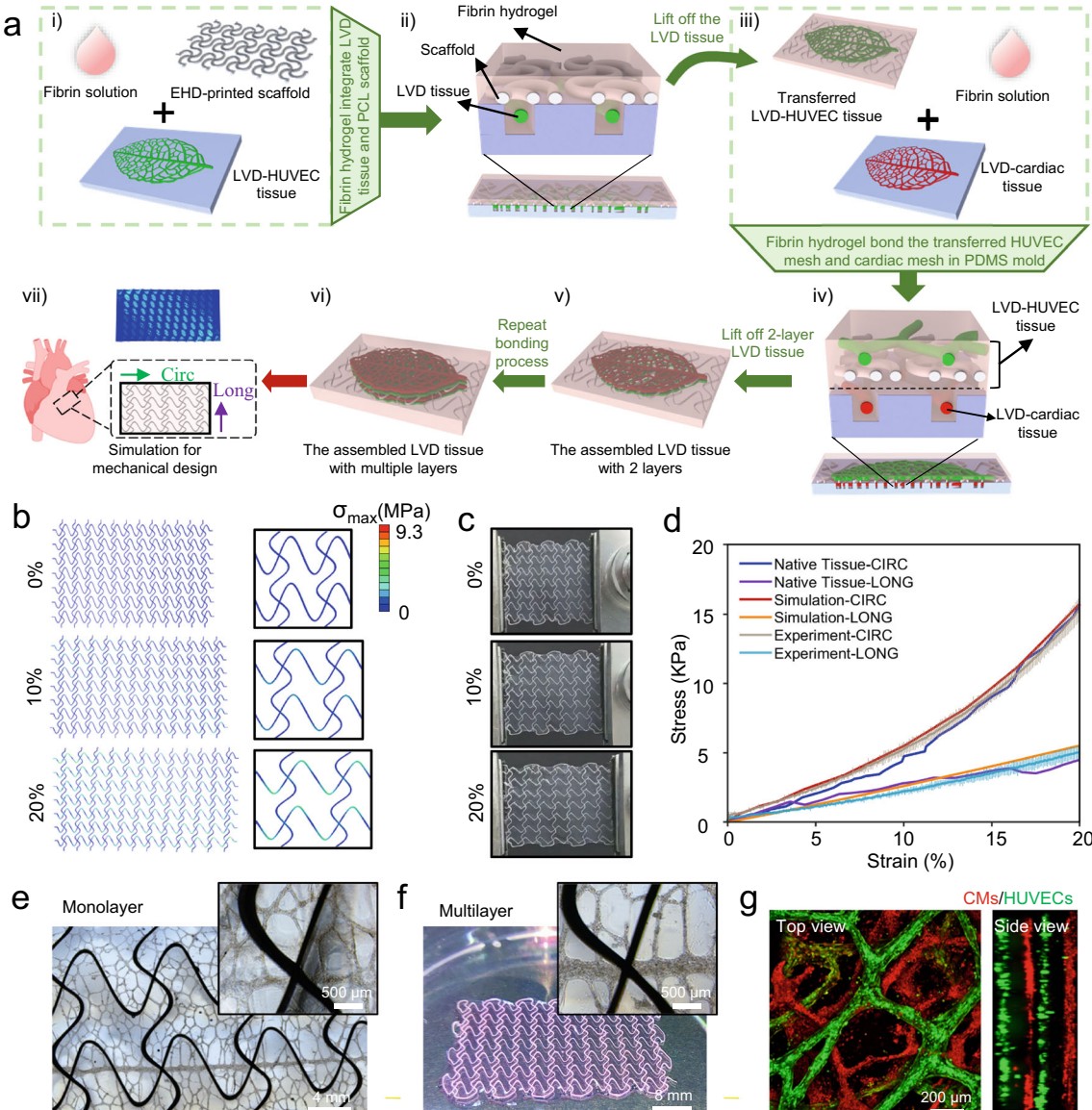

**Fig. 6 | Engineering of 3D pre-vascularized LVD-cardiac tissue constructs with the programmed mechanical property. a** The transfer and assembly process of the multiple LVD tissues. FEA simulations (**b**) and macroscopic images (**c**) of the 3D tissue composite upon uniaxial stretching (from top to bottom: 0%, 10%, and 20%) along the CIRC direction. **d** The stress–strain curves along the CIRC and LONG direction among the corresponding FEA calculations, experimental measurement, and adult rat right ventricular myocardium. $n = 3$ independent samples. **e** Bright-field of the transferred monolayer LVD-rat-cardiac tissue. The macroscopic (**f**) and confocal (**g**) images of the assembled 4-layer LVD-rat-cardiac construct, consisting of red LVD-rat-cardiac and green LVD-HUVEC tissues superimposed with each other.

scaffold and fibrin hydrogel, subjected to out-of-plane bending during the rolling process (Fig. 7f). The calculated results confirmed that the maximum strain force in the scaffold was ten thousand higher than the hydrogel with different wrapping angles (Fig. 7g). The empirical results of live-dead staining show no significant differences between the percentage of viable cells before and after the injection of the assembled cardiac tissues in vitro, suggesting that the rolling-induced large compressive deformation and flow-induced recovery process did not negatively influence the 3D LVD tissues (Fig. 7h, i).

## Discussion

Myocardial infarction is a leading cause of morbidity and mortality worldwide. Intensive efforts have been invested in treating myocardial infarction by injecting stem cells or CMs directly into the myocardial wall. Unfortunately, the weight of evidence so far indicates no clinically meaningful benefit resulting from injection therapies, owing to the limited engraftment capacity of the injected CMs and the lack of control over the cells' survival, arrangement, and electromechanical coupling[48]. Recently, biomaterial-based engineered tissue constructs have been investigated for their therapeutic potential. Some tissue constructs, such as hiPSC-CMs-laden fibrin hydrogel, have been implanted to treat severe heart failure in humans, showing some therapeutic benefit compared with the injection of cells[49,50]. However, because the engineered constructs do not recapitulate the electro-physiological and architectural features of the native myocardium, their ability to integrate functionally with the host myocardium is hampered. In addition, to achieve optimal performance and reduce the risk of arrhythmogenesis, the engineered tissues need to have a mature phenotype manifested in characteristic gene-expression profiles, an elongated and aligned morphology, organized sarcomeres, axial electrical connectivity via gap junctions, and matured action potential and calcium transients[3,51]. Therefore, achieving a controlled,

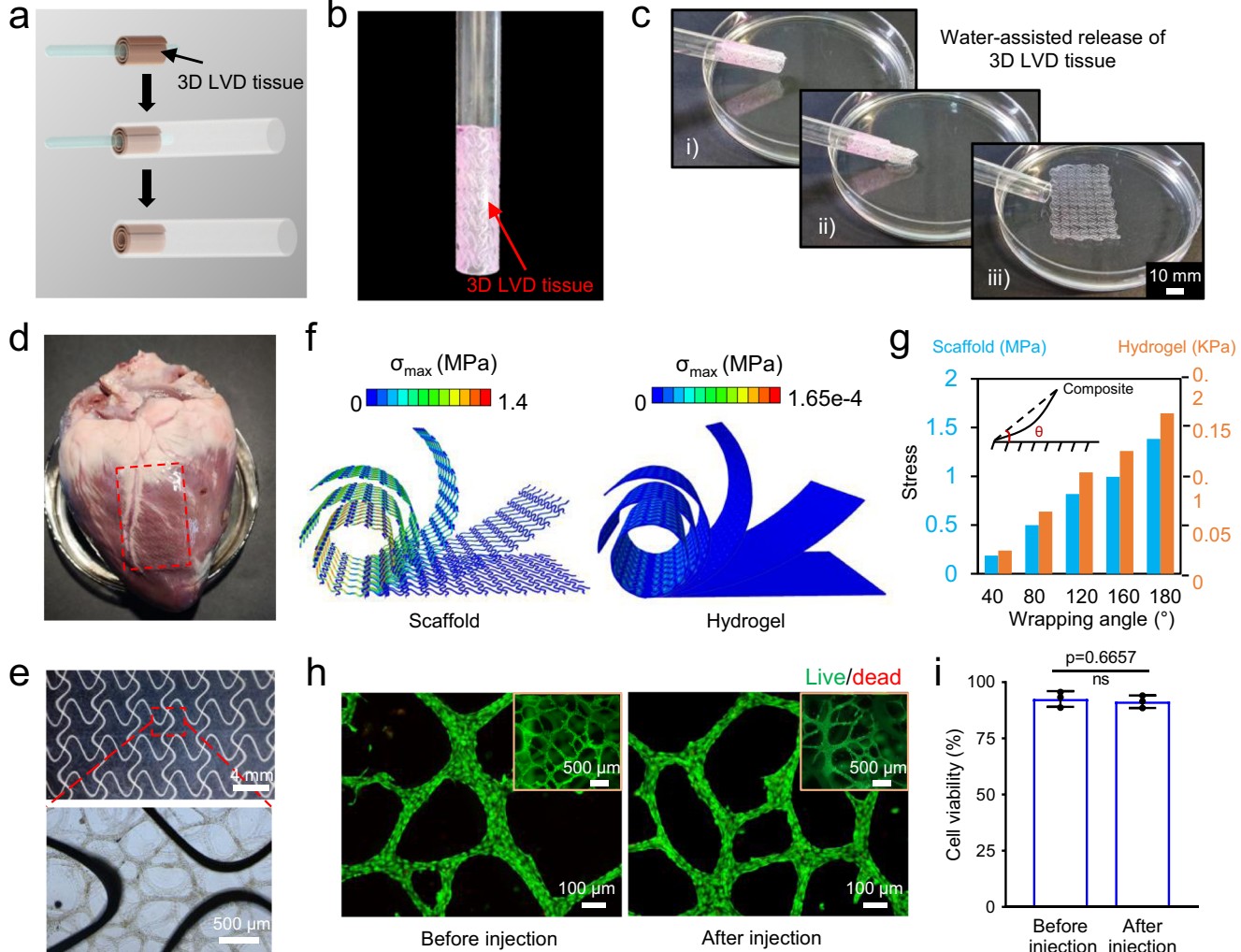

**Fig. 7 | Injectable delivery of 3D LVD-cardiac tissue constructs. a** Schematic of rolling up cardiac construct. **b** The rolled cardiac construct in the tubing. **c** In vitro injectability and shape recovery of the cardiac construct. **d** Injected cardiac construct on porcine heart. **e** The structural integrity of the injected cardiac construct. **f** The simulation results of the strain force distribution within the scaffold and fibrin hydrogel of the LVD-cardiac construct during the rolling process. **g** Quantification of the maximum strain force within the scaffold and fibrin hydrogel respectively, when the LVD-cardiac constructs were wrapped in different angles. **h** Fluorescence imaging of live (green) and dead cells (red) in LVD tissues before and after injection. Inset: lower-magnification image of LVD tissues. **i** Cell viability of cardiac constructs before and after injection. $n = 3$ independent samples. Data are represented as mean values ± standard deviation. The significant difference is determined by unpaired two-tailed *t*-test, 95% confidence interval. ns ($p > 0.05$) no significant difference.

biomimetic anisotropic organization of matured CMs is crucial to improve the utility and functionality of 3D cardiac tissue constructs.

In this study, leaf-venation-inspired microchannels were employed to direct the formation of functional cardiac tissues and the hierarchical pre-vascular networks, considering its merits for efficient mass transfer of fluid and electrical signals and damage tolerance[29,52–54]. In plants, the leaf veins, consisting of densely packed cells and aligned fibers, allow the transmission of action potentials, which progression was characterized by the rapid propagation of local intracellular $Ca^{2+}$ concentration[53,54]. This process resembles the electromechanical coupling in the mammalian myocardium: cardiac mechanical contraction is triggered by electrical activation via increased cytoplasmic $Ca^{2+}$ concentration[54]. Inspired by their similarity, we tend to employ the leaf-venation networks for directing the formation of cardiac tissues with interconnected close-packed populations, enabling the robust electrophysiological activity of $Ca^{2+}$-based electrical transmission. Besides, considering the significant similarity to mammalian vasculature regarding the hierarchical vascular structure and high-efficient transportation function, leaf venation is an ideal template for generating biomimetic cardiac vasculature systems[27].

The presented LVD strategy enables the culture and maturation of rat CMs and hiPSC-CMs within fibrin hydrogel, resulting in the formation of large-scale physiologically relevant cardiac tissues. The engineered LVD-cardiac tissues are tubular structures with highly aligned and densely packed populations along the heavily compacted fibrin hydrogel. There are several crucial merits to this structure. First, analogous to the native CMs that are organized into spatially well-defined cardiac bundles, LVD-cardiac tissues are in highly anisotropic architectures with dense populations, where each CM adjoins neighboring CMs by specialized intracellular junctions, benefiting their synergistic electrophysiological activity. Second, the CMs are mainly located on the tubular structures' outer surface, thus satisfying their high demand for mass transfer. Third, the heavily compacted fibrin fibers provide the appropriate cell-binding sites and resemble the aligned structure and stiffness of native ECM[37]. Hence, all these biomimetic features contribute to the improved maturation of the engineered LVD-cardiac tissues, which exhibits elongated architecture, well-defined sarcomeric striations, macroscopically synchronous contractions, detectable electrophysiological activity and upregulation of several crucial maturation genes.

We have demonstrated the feasibility of the LVD system with built-in conductive microelectrodes to provide electrical stimulation for the cultured rat CMs and further monitor their extracellular field potentials in situ. It is of great significance to conduct an in-depth study of culturing hiPSC-CMs in the microelectrode-integrated LVD system in the future, considering their potential as an attractive cell source for drug development and regenerative therapy applications. Besides the environmental cues, there are other methods to improve the maturation of hiPSC-CMs, such as electrical/mechanical stimulation and long-term culture with evolving media supplementation[55,56]. As a non-invasive approach, our microelectrode-integrated LVD system would have the superiority of culturing matured hiPSC-CMs-based cardiac tissues with functionality. One concern is the limited distribution of microelectrodes in the LVD system. In this work, we just inserted two microelectrodes in the primary channel of each PDMS substrate as a demonstration. In our experience, the platinum wires could be successfully inserted into the PDMS substrate's primary channel by hand. However, it is challenging to insert the platinum wires exactly into the branch channels, whose widths were mostly less than 150 μm. Some specialized positioning devices should be developed to assist the operation. Moreover, the effect of the 30-μm-diameter microelectrodes on the growth of the cellular bundles in the branch channels still needs further investigation since their final diameter is mostly less than 50 μm.

Vascularization of engineered tissues is of the essence for cell survival and function, especially for CMs with a high-metabolic rate. To facilitate an efficient mass transfer, all the CM bundles in the native myocardium are wrapped around by the dense networks of capillaries, whose functionality are relied heavily on their tree-like hierarchical branching structure[7]. Early attempts to create vascularized cardiac tissues through the simple addition of vascular endothelial cells showed that some level of spontaneous-assembly of tubular structures in ECM occurs; however, these resultant vascular components are usually random, uncontrollable, and less efficient for facilitating in vivo regeneration and anastomosis of perfusable blood vessels[57]. Although perfusable vascularized structures can be engineered by seeding and pre-organizing endothelial cells along the inner surfaces of the defined microfluidic channels, it remains challenging to fabricate endothelialized channels with biomimicking hierarchical structures at the microvasculature size scale (<30 μm), as well as to integrate them with the functional cardiac tissues[58,59]. Another potential strategy for engineering vasculature is seeding cell/matrix mixtures into grooved templates, guiding the self-assembly of cell/matrix mixtures into the tubular structure with specific patterns[33]. Previous works indicate that such self-assembled tubular structures with a prescribed geometry in vitro can provide a template that defines the neovascular architecture in vivo on implantation and enhance anastomosis and vascular functionality[32]. Parallel arrays of cords resulted in a similar capillary network, while a single cord with a bifurcation resulted in a perfused branch point. However, the described tubular structures are in simple patterns that tend to shrink during the self-assembly process. In this study, we attempt to employ the LVD strategy to engineer large-scale, highly interconnected, and hierarchical microvascular tissues that capture the complexity of the native EC arrangement in tubular branching structures. Considering that previous work has demonstrated the ability of the tubular vasculature to template the neovascular architecture in vivo, we assume that the LVD-HUVEC tissue can guide the formation of neo-vasculature with the biomimetic hierarchical architecture and induce ingrowth and anastomoses of host vasculature in vivo which can facilitate the mass transfer. Therefore, a large and separate animal study needs to be conducted in the future to investigate the growth of LVD-HUVEC tissues and their effect on the long-term survivability of the LVD-cardiac tissues post-implant.

The mechanical integration of the LVD-cardiac tissues into the host tissues also needs further investigation through animal study.

Previous works have demonstrated the benefits of mechanical support from biomaterial-based cardiac patches, which enable to reduce myocardium wall stress and subsequently limit adverse remodeling and improve cardiac function after myocardial infarction[60]. However, the optimum patch design is still unknown. In our study, EHD printing was employed to produce the elastic PCL scaffold with precise microscale serpentine structures, thus granting us freedom in tailoring the mechanical property of the LVD-cardiac tissue constructs. This allows us to study the effect of cardiac patches with programmed anisotropic mechanical properties on the heart's remodeling process and pumping function. For this purpose, the LVD-cardiac tissue constructs should be implanted onto the hearts of large animal models, such as pigs and dogs.

The materials component is another crucial consideration of the engineered cardiac tissues, since the biomaterials allowed for clinical implantation are quite limited. Recently, significant efforts have been devoted to enhancing the electrical conductivity of the cardiac construct, which has shown merits in facilitating CM functionalization, coupling, and local synchronous contraction[61]. However, adding conductive components (e.g., carbon nanotubes) into the scaffold or ECM in those works would make the engineered cardiac construct challenging to use in clinical applications. In our LVD tissues, only two biocompatible biomaterials are involved: fibrin, a typical naturally derived hydrogel, and PCL, a synthetic polymer broadly used in the U.S. Food and Drug Administration (FDA)-approved implants. Fibrin hydrogel provides abundant cell attachment sites that facilitate cell spreading and function and enable macroscopic tissue compactions that yield high cell density and cellular alignment[62]. EHD-printed PCL fiber scaffold was employed as the mechanical support to achieve the transfer of LVD tissues from the template, the assembly of LVD tissues into 3D tissue composites with anisotropic mechanical properties that conform to the nonlinear deformation of the native myocardium, as well as the rolling and release of the assembled LVD tissues for minimally invasive implantation. Such a combination of synthetic structural materials (such as PCL and poly-(lactic-co-glycolic acid)) with natural materials (such as fibrin and collagen) might be a prevalent trend in engineering cardiac tissues, whose functionality is highly related to the ECM microenvironment and mechanical property[3,63,64].

We adopted the bottom-up strategy to assemble the LVD tissues into 3D pre-vascularized cardiac tissues. The advantages of this strategy are as follows: (i) allowing for the assembly of multiple tissue components, such as cellular layers and scaffolds with predefined structural and mechanical properties, (ii) preculturing functional modular tissue under the different culturing systems, followed by assembly into 3D tissues with immediate functionality on demand which allowed for immediate implantation, (ii) eliminating the need for a complicated system for in vitro culture of thick tissues, as each thin tissue mesh can be cultured separately without oxygen deficiencies. By employing this strategy, we can engineer the pre-vascularized cardiac constructs with anisotropic architectural and electrophysiological features of the native myocardium. Other engineer strategies, such as bioprinting, enable spatially defined multiple cell/hydrogel positioning; however, they face many challenges in engineering such functional cardiac tissues for clinical application. For example, CMs are rounded and do not form interconnected syncytium immediately upon tissue fabrication, and many days may be required for the cells to elongate and connect so that they can exhibit a synchronous contractile function. The predefined structure might be destroyed owing to the cell-spreading-induced force and a complicated perfusion system with different culture mediums should be needed to engineer the cardiac tissue with multiple types of cells.

In a word, the described LVD strategy enables the self-assembly of cell-laden ECM hydrogels into large-scale, highly aligned and densely packed tissues, including hierarchical pre-vascular and cardiac tissue. Especially, clinically relevant and functional cardiac constructs derived

from rat CMs and hiPSC-CMs were successfully engineered that resemble the anisotropic architectural and electrophysiological features of the native myocardium. This was evidenced by elongated and aligned morphology, well-defined sarcomeric striations, macroscopically synchronous contractions, accelerated calcium transients, and detectable electrophysiological activity, as well as the upregulation of several crucial maturation genes. With the EHD-printed PCL fiber scaffold, we successfully fabricated the 3D pre-vascularized cardiac constructs with anisotropic mechanical properties that conform to the nonlinear deformation of the native myocardium and show their feasibility of minimally invasive implantation. In the future, we expect to engineer the functional LVD-hiPSC-derived-cardiac constructs in vitro and conduct large-scale and long-term animal studies to investigate their post-implant survivability and mechanical integration into the host tissues. We envision that this versatile and scalable tissue engineering strategy can contribute to the ultimate goal of engineering aligned, vascularized, and mechanically programmed functional cardiac tissues for clinical therapy.

## Methods

### Materials

This study did not generate new unique reagents. Fibrinogen from bovine plasma (341573) was purchased from Sigma-Aldrich (USA). Thrombin from bovine plasma (T2081) was bought from Solarbio (China). Fibrinogen and thrombin were separately dissolved in Dulbecco's phosphate-buffered saline (DPBS) (Gibco, USA) and filtered to obtain a desirable concentration solution for further cell culture. Fibrinogen from human plasma, Alexa Fluor™ 488 Conjugate (F13191) was purchased from Thermo Fisher Scientific (USA) and a 1.5 mg/ml stock fluorescent fibrinogen solution was prepared according to the instructions. Endothelial growth medium (EGM-1, 1001) and endothelial growth supplements were obtained from ScienCell (USA). Collagenase type ‖ (17101015) was purchased from Thermo Fisher Scientific (USA). 5-Bromo-2'-deoxy-uridine (5-BrdU, B5002) was purchased from Sigma-Aldrich (USA) and was dissolved in serum-free Dulbecco's modified Eagle's medium/Nutrient Mixture F-12 (DMEM/F12, Gibco) to obtain a 10 mM concentration stock. The Live/Dead™ viability/cytotoxicity kit (L3224), red and green cell tracker staining kit (C7025, C34552) and the cytoskeleton staining kit of Alexa Fluor 594 phalloidin (A12381) were purchased from Invitrogen (USA). 4′, 6-diamidino-2-phenylindole (DAPI) was purchased from Beijing Dingguo Changsheng Biotechnology Co., Ltd (China). Fluo-8 AM kit (21082) was purchased from AAT Bioquest (USA). The primary antibodies of mouse anti-sarcomeric α-actinin (ab9465), rabbit anti-CX43 (ab11370) and rabbit anti-CD31 (ab182981) were purchased from Abcam (USA). Rabbit anti-cTnt (15513-1-AP) was purchased from Proteintech (China). The secondary antibodies of Alexa Fluor 488 goat anti-mouse IgG (A11029) for α-actinin as well as Alexa Fluor 594 goat anti-rabbit IgG (A11012) for CX43 and CD31 were purchased from Invitrogen (USA). Goat anti-rabbit IgG (Alexa Flour-488) secondary antibody (150081) for cTnT and CD31 was purchased from Abcam (USA). RNAfast200 kit (220010) was purchased from Fastagen (China). PrimeScriptTM RT reagent Kit (RR047A) and TB Green Premix Ex TaqTM II (RR820L) were obtained from TAKARA (Japan). Fetal bovine serum (FBS), DPBS, PBS, trypsin-EDTA and other relevant reagents were purchased from Gibco (USA) unless otherwise mentioned.

### Cell culture

HUVECs (DFSC-EC-1) were purchased from Shanghai Zhongqiaoxinzhou Biotech Co., Ltd (Shanghai, China), and maintained in EGM-1 medium supplemented with 10% (v/v) FBS and endothelial growth supplements. HUVECs were used at passages 3−6. Neonatal rat CMs were isolated from the 2-day-old Sprague-Dawley rats without considering gender differences. The rats were purchased from Laboratory Animal Center of Xi'an Jiaotong University and -30 rats were used for

the cell isolation. All procedures were performed according to the guide for the care and use of laboratory animals at Xi'an Jiaotong University and were approved by Animal Ethics Committee at Xi'an Jiaotong University (Approval numbers: 2021-1242). Briefly, the rat hearts were separated and cut into pieces (-1 mm²). Then, they were dissociated in PBS containing 1.5 mg/mL collagenase type ‖ and performed digestion by continually shaking in a 37 °C water bath shaker at 100−200 rpm. The digestion procedure was repeated 7−8 times for 5 min each time, and the supernatant was collected into DMEM/F12 medium containing 10% FBS. When the heart tissue was completely digested, the supernatant was centrifuged at $220 \times g$ for 5 min and resuspended in DMEM /F12 medium with 10% FBS and 0.1 mM 5-BrdU. After incubation for 1.5 h to separate the fibroblasts from rat CMs, the cell suspension was transferred to fibrin-coated cell culture dishes at an appropriate density. The rat CMs were cultured for 48 h and washed three times with PBS before use. hiPSC-CMs (HELP4111, NovoCell™) were purchased from Help Therapeutics (Nanjing, China) and cultured according to manufacturer's instructions. Briefly, a six-well cell culture plate was added with coating solution (F00201, Help Therapeutics) of 1 ml in a single well and placed in a 37 °C incubator. After 1 h, the coating solution was removed and changed to 1 ml cell thawing medium (F00901, Help Therapeutics) preheated at room temperature for 30 min. According to manufacturer, NovoCell™ hiPSC-CMs were produced by differentiating hiPSCs for 18 days, and consisted mainly of ventricular cells with autonomous electrophysiological activity. Cryovials of hiPSC-CMs were quickly thawed in a 37 °C water bath, transferred into a 15 ml centrifuge tube, and added with a 10 ml preheated cell thawing medium. The hiPSC-CMs suspension was then centrifuged at $300 \times g$ for 5 min, resuspended with cell thawing medium, added to the pre-coated six-well plate with $2 \times 10^6$ cells in a single well, and cultured in cardiac maintenance medium (F00301, Help Therapeutics). All culture media contained 1% penicillin−streptomycin. After 3 days of culture, cells were dissociated for engineering LVD tissue.

### Fabrication of leaf-venation-inspired microchannel substrate

The leaf-venation-inspired microchannel substrates were fabricated using standard soft lithography as previously described[29]. Briefly, a photomask of the leaf-venation network was produced by outsourcing based on the established digital model. A 4-inch silicon wafer was spin-coated with EPG535 photoresist (Everlight chemical industrial Co., Taiwan) at 1500 rpm for 40 s. Upon solidification, the coated silicon wafer was exposed to ultraviolet light for 6 s, and the exposed region was dissolved in 0.5% NaOH solution. Next, the surface-patterned silicon wafer was dried-etched in an inductively coupled plasma etching machine (Oxford, ICP180) to achieve a depth of 300 μm. The resultant wafer was coated with octafluorocyclobutane to facilitate subsequent demolding. Then, A mixture of Sylgard 184 silicone elastomer components at a weight ratio of 10:1 was degassed and poured onto the silicon wafer, curd at 75 °C for 4 h, and peeled off to obtain a PDMS layer with the negative pattern of the leaf-venation-inspired microchannel. This PDMS layer was trimmed into a rectangular shape, coated with octafluorocyclobutane, and then used as a negative mold to produce the final PDMS leaf-venation-inspired microchannel substrate with a chamber by repeating the PDMS-replication process as described above (Supplementary Fig. 13).

### Formation and culture of LVD tissues

To generate LVD tissues, the leaf-venation-inspired microchannel substrate was autoclaved, treated with plasma for 3 min, immersed in 1% (wt/vol) pluronic F-127 solution for 1 h to prevent cell attachment and rinsed with PBS for 10 min just prior to use. To prepare cells for seeding, HUVECs were trypsinized with 0.05% trypsin-EDTA for 1−2 min and centrifuged at $350 \times g$ for 5 min; rat CMs were trypsinized with 0.25% trypsin-EDTA for 1-2 min and centrifuged at $220 \times g$ for

5 min; hiPSC-CMs were trypsinized with specified trypsin (F00101, Help Therapeutics) for 3–5 min and centrifuged at $300 \times g$ for 5 min. The obtained cells (HUVECs/rat CMs/hiPSC-CMs) were suspended in 6 U/ml thrombin solution and then mixed with the 6 mg/ml fibrinogen solution in a volume ratio of 1:1 to prepare the cell-laden fibrinogen suspension. The final concentration of fibrinogen in the suspension was 3 mg/ml, and cell density in all suspensions was fixed at $2 \times 10^7$ cells/ml. A 120 µL cell-laden suspension was quickly added over the center of each leaf-venation-inspired microchannel substrate, which was placed in a cell culture dish with a diameter of 10 cm. While most cell-laden suspensions would automatically disperse and fill all the microchannels, some excess cell suspensions were scraped off with a cover glass. After incubation at 37 °C for 15 min to allow fibrinogen gelation, 5 mL culture medium was added into the chamber of the PDMS substrate (Supplementary Fig. 14) and refreshed every 2 days. In addition, to generate non-aligned tissues as control, the cell-laden fibrinogen suspension was added into the leaf-venation-inspired microchannels after the 3-min oxygen plasma treatment, followed by the same gelation and culture procedure. The developing process of HUVEC/fibrin hydrogel in the leaf-venation-inspired microchannel was recorded using an inverted microscope (Nikon Ti-S, Japan) at the same position of the microchannel. The beating videos of the hiPSC-CM LVD constructs were recorded by the inverted microscope.

HUVEC cells pre-labeled with a red cell tracker were encapsulated into fluorescent fibrinogen suspension to characterize the dynamic formation process of tubular structures in LVD tissues. The fluorescent fibrinogen solution at the concentration of 6 mg/ml was prepared by mixing 7.5 mg/ml normal fibrinogen and 1.5 mg/ml Alexa Fluor™ 488 Conjugated fibrinogen in a volume ratio of 3:1. Following the same gelation and culture procedure as described above, the fluorescent HUVECs/fibrin hydrogel was characterized with a laser scanning confocal microscope (LSCM, Nikon A1) at different time points.

### Electrophysiologic assessment of LVD-cardiac tissues

A custom-built microelectrode-integrated device was produced for culturing LVD-cardiac tissue and their in situ biologically relevant electrophysiological assessment. Specifically, the thin platinum wires with an average diameter of 30 µm were placed in a hollow needle and inserted into the primary channel of the as-prepared PDMS leaf-venation-inspired microchannel substrate by hand. The hollow needle was then removed, leaving the thin platinum wire passing through the PDMS layer. The end of the integrated thin platinum wires with a length of ~1 mm was exposed to the cells in primary channels, working as microelectrodes to acquire their electrical activity. After that, the PDMS elastomer was coated onto the bottom of the PDMS leaf-venation-inspired microchannel substrate to seal the platinum interconnects. The conductive silver paste was pipetted at the contact pad to connect the platinum to the electrophysiological measurement setup. The obtained microelectrode-integrated PDMS substrates could be autoclaved and stored for subsequent cell culture. Each PDMS substrate was integrated with two microelectrodes in the primary channel in this work. A platinum electrode, as an internal reference, was placed in the culture medium during the signal recording. The obtained electrical signal was generated by the sum of the synchronized action potentials of all CMs.

The biomedical instrumentation for recording the electrophysiological signals was a controller board (RHS2000, Intan Technologies, Los Angeles, USA) and amplifier chips (RHS2116, Intan Technologies, Los Angeles, USA). A printed circuit board (PCB) was designed using the KiCad software. The PCB contained two connectors (A79024, Omnetics connector corporation) and 60 spring probe electrodes (POGO PIN, SZXHN, Shenzhen, China) to connect the contact pads and the amplifier chips. The PCB board was placed in a Faraday cage to screen the extra noise during the recording process. The data acquisition system was controlled by Intan stimulation/recording controller software. The recorded data was processed in MATLAB (The Mathworks, USA) with an open-source m-code function provided by Intan Technologies and further analyzed using Origin (OriginLab Corporation, USA).

### EHD printing of elastic PCL scaffold

To lift LVD tissues out of the PDMS substrate, an elastic PCL scaffold was produced by a custom-built melt-based electrohydrodynamic printing platform according to the predesigned structures. Medical-grade PCL (Jinan Daigang Biomaterial Co., Ltd, China) with a molecular weight of 80,000 was selected as the electrohydrodynamic printing biopolymer due to its good biocompatibility. To print the PCL scaffold with a filament width of 0.1 mm, the process parameters of applied voltage, nozzle-to-collector distance, feeding rate, stage moving speed, and melting temperature were fixed at 2.7 kV, 2 mm, 45 µL/h, 5 mm/s, and 80 °C respectively. The actual spacing and height of the electrohydrodynamically printed scaffolds were measured with optical microscopy (Ti-S, Nikon). The microscopic morphology of the printed scaffolds was further characterized with a SEM (SU8010, Hitachi)

### Immunofluorescence staining and microscopy

LVD-HUVEC tissues cultured for 3 days in vitro were washed twice with PBS and fixed with 4% paraformaldehyde for 30 min at room temperature. After being rinsed with PBS three times, samples were incubated for 1 h in a solution consisting of 3% bovine serum albumin and 1% Triton X-100 on a rocker for membrane permeabilization and blocking. Human CD31 was stained with rabbit anti-CD31 primary antibody (1:50) in a 1% BSA solution at 4 °C overnight, followed by goat anti-rabbit IgG (Alexa Flour-488) secondary antibody (1:1000) in a 1% BSA solution in the dark for 1.5 h on a rocker. Subsequently, samples were stained with phalloidin (1:500) for F-actin (cytoskeleton) and DAPI for nuclei. LVD-cardiac tissues from rat CMs and hiPSC-CMs cultured for 3 to 17 days in vitro were washed twice with PBS and fixed with 4% paraformaldehyde for 30 min at room temperature. After being rinsed with PBS three times, samples were permeabilized with 0.25% Triton X-100 in PBS for 20 min and then blocked with 10% goat serum in DPBS for 1.5 h on a rocker. Subsequently, samples were incubated with the primary antibodies of mouse anti-α-actinin (1:200) and rabbit anti-CX-43 (1:200), and rabbit anti-cTnT (1:200) respectively at 4 °C overnight. After being washed with DPBS three times, goat anti-mouse IgG (Alexa Flour-488) secondary antibody (1:1000) for α-actinin and goat anti-rabbit IgG (Alexa Fluor 594) secondary antibody (1:1000) for CX43 secondary antibodies, goat anti-rabbit IgG (Alexa Flour-488) secondary antibody (1:1000) for cTnT were added respectively and incubated for 1.5 h in the dark at room temperature on a rocker.

The images were obtained using a laser scanning confocal microscope (Nikon A1). When scanning samples with the same cells, fluorescence images were acquired in spatial sequence using equal laser intensity and detector gain and adjusted for contrast and brightness using ImageJ.

### SEM and TEM characterization

For SEM, the LVD tissues were fixed with 4% paraformaldehyde, dehydrated with graded alcohol aqueous solutions, freeze-dried, and imaged with SEM (SU8010, Hitachi). For TEM, the LVD tissues were peeled off from the leaf-venation-inspired microchannel substrate, fixed in 3% glutaraldehyde at 4 °C overnight, and processed for electron microscopy. Ultra-thin sections were cut with an ultra-microtome (Ultracut, Reichert-Jung) and observed under a transmission electron microscope (TEM300, Itachi) operated at 75 kV.

### Atomic force microscopy

The Young's modulus of LVD tissues was measured using AFM (NanoWizard 4 XP BioScience, Bruker). Measurements were collected in contact mode using AFM cantilevers (NP-O10, Bruker), modified by

gluing glass beads (12 μm, Sigma) to the cantilever underside. The nominal spring constant of the cantilever was 0.06 N m$^{-1}$, calibrated by the thermal fluctuation method. Prior to indentation tests, the sensitivity of the cantilever was set by measuring the slope of force-distance curves acquired on the petri dish. Three force maps were collected in separate areas on each sample and averaged for the sample value with no distinction between primary and branch channels.

## Calcium transient and beat rate analysis

The Ca$^{2+}$ transients of rat CMs seeded in microchannel were evaluated using a Fluo-8 AM kit according to the manufacturer's instructions. The LVD-cardiac samples were washed with DPBS three times and then incubated with a working solution (5 μmol Fluo-8 AM and 0.4 mg/ml Pluronic F-127 in serum-free DMEM/F12 medium) for 60 min at 37 °C. The working solution was removed and changed to serum-free DMEM/F12 medium. The fluorescence images of the samples were captured by a high-speed sCMOS camera (ZYLA 5.5, Andor) at a sampling rate of 50 Hz. The fluorescence signal during cell contractions was evaluated with Image J. The measured fluorescence intensity in arbitrary units was plotted against time.

The spontaneous beating rate of LVD and control tissues was determined optically by counting the number of beats per minute in the bright-field mode of an inverted light microscope.

## Viability assay

According to the manufacturer's instructions, cell viability was assessed using Live/DeadTM viability/cytotoxicity kit. After culture, the samples were washed with PBS three times, followed by incubation in the staining solution for 20 min at 37 °C in the dark. Calcein acetoxymethyl (Calcein-AM) produced a uniform green fluorescence in live cells, while ethidium homodimer-1 produced a bright red fluorescence in dead cells. The samples were washed three times with DPBS and viewed under LSCM (Nikon A1). Cell viability was quantified using ImageJ.

## qRT-PCR

LVD-cardiac tissues from rat CMs and hiPSC-CMs were homogenized to obtain mRNA using an RNAfast200 kit, which was then converted to cDNA using PrimeScriptTM RT reagent Kit. qRT-PCR was performed on an ABI 7500 RT-PCR system (Applied Biosystems, USA) using the mixture of the obtained cDNA and TB Green Premix Ex TaqTM II according to the manufacturer's instructions. The primer sequences used in this study are listed in Supplementary Table 1. The expression of target genes, normalized to that of the target genes in the control group, was quantified by the ΔΔCt method using GAPDH as the housekeeping gene.

## Image analysis

The orientation of cells and fibrin hydrogel fibers was quantified using OrientationJ plug-in function of ImageJ software. Cell orientation was quantitatively evaluated based on the direction of the nuclear shape index of DAPI-stained nuclei, as previously demonstrated[17]. Quantitative data of the nuclear alignment angles, defined as the orientation of the major elliptic axis of individual nuclei with respect to the horizontal axis, were obtained. All nuclear/fiber alignment angles were normalized to a dominant direction, defined as the mean orientation of all nuclei/fiber.

The width and cross-sectional area of LVD-HUVEC tissues, size of CX43, and length of the sarcomere, were quantified in ImageJ. Specifically, the bright-field images of HUVEC/fibrin hydrogel during the compaction and tissue remodeling process were imported into ImageJ to measure their width. The confocal fluorescence images of the composite of cell tracker-labeled cells and FITC-label fibrin hydrogel were reconstructed to 3D profiles in NIS Viewer software (Nikon), and their cross-sectional images were exported. The area of cells/fibrin hydrogel in the cross-sectional images was measured in ImageJ and normalized by the cross-sectional size of corresponding primary or

branch channels to obtain the area fraction of cells/fibrin hydrogel. The area of CX43 was measured in ImageJ and normalized by the size of corresponding primary or branch channels to obtain the area fraction of CX43.

Bright-field videos were analyzed for beating physiology using an open-source motion tracking software (available at https://huebschlab.wustl.edu/resources/)[65,66]. The videos were recorded at 100 frames per second using an inverted microscope (Nikon Ti-S, Japan) and exported as a series of single-frame image files. These image files were then imported into the motion tracking software, which can calculate the contractile motion of tissues via an optical flow algorithm that compared the position of 8×8 pixel macroblocks at frame i to their position at frame i + 5 (corresponding to the motion in 50 ms). The software can automatically output the motion vectors, maximal contraction velocity and motion heat map.

## Drug treatment

Isoproterenol (Sigma-Aldrich, USA, I5627) was dissolved in distilled water to make stock solutions, and dilution was made in serum-free culture medium DMEM/F12 (1:1, Hyclone, USA) to the required concentration. For drug treatment, 5 μM isoproterenol was pipetted into the culture medium.

## Finite-element analysis

FEA was adopted by employing commercial software ABAQUS (SIMULIA, Providence RI) to analyze deformations and stress–strain responses for the composite of elastic scaffold and fibrin hydrogel. The ten-node linear tetrahedral and eight-node hexahedral hybrid elements were used for the scaffolds and hydrogel, respectively. One of the considerable aspects of FEA analysis is the dependency of its result accuracy on the mesh size since the FEA model with coarse mesh is not representative of the continuous model and leads to deviation from exact results. Thus, a mesh sensitivity analysis was performed by constantly decreasing the mesh size (or increasing the number of elements) to reach mesh size-independent results. After completing this analysis, a suitable mesh size on the FEA models was obtained for the following FEA analysis.

This study used the model simulation of linear elasticity for PCL, and the material properties were assumed isotropic with a Poisson's ratio of 0.3 and Young's modulus of 300 MPa, which was determined from compression testing of solid gage pure PCL specimens[67]. For hydrogels, a hyper-elastic constitutive relationship following the Mooney-Rivlin law was usually used in finite-element analysis. However, it noted that the stress–strain curves of hydrogels had been shown in previous studies to be approximately linear within small deformations[68,69]. Thus, linear elastic constitutive relations were used in this study to improve the computational efficiency with the elastic modulus and Poisson's ratio to be 0.5 KPa and 0.25[70], respectively.

Tensile simulation analysis of the composite model body was used to simulate static uniaxial tensile testing by applying a 20% displacement constraint to the top of the composite model and fixing its bottom as a boundary condition. The tensile reaction force was obtained for the composite in the post-processing stage. Then, the modulus of elasticity, $E_{FEM}$ was obtained from the following equation:

$$E_{FEA} = \frac{F_{FEA}}{A \times \varepsilon_{FEA}}$$

In which, $F_{FEA}$ is tensile force at the end of the analysis, $A$ is the cross-sectional area of the composite in tension, and $\varepsilon_{FEA}$ is a compressive strain of 20%.

The bending deformation of the composite was simulated using the same material analysis parameters as the tensile process. With the left side of the composite fixed, a primary reference point coupling the whole surface was established on the right side, and an angular

displacement load was applied to the reference point. The stress–strain nephograms of the scaffolds and hydrogel were output, respectively.

The simulation analysis of the composite in this study considers the interaction between the two models through the 'tie contact' in ABAQUS.

## Mechanical analysis

Uniaxial tensile tests were performed on the 3D composites of EHD-printed scaffold and fibrin hydrogel (30 mm × 30 mm) using an electromechanical universal testing machine (103 A, Wance, China). The composites were fixed on the testing plate using double-sided adhesive tape and stretched using a load cell of 50 N at a crosshead speed of 1 mm min$^{-1}$ at room temperature. A digital camera (D7100, Nikon, Japan) was used to keep track of observation on the whole tensile responses. The composites were tested in both horizontal and vertical directions according to the x- and y-direction of the printed plane. Three samples were tested for each composite configuration, and the effective modulus was determined by the slope of the linear stage of the stress–strain curve.

## Statistics and reproducibility

All the statistical analysis was performed using Origin 2020 (OriginLab Co, Northampton, MA) and Prism 9 (GraphPad, Inc.) software. All quantitative results were presented as mean ± standard deviation. Statistical significance was determined using unpaired two-tailed Student's t-test between two groups. The differences were considered statistically significant if the value of p was <0.05 (∗). All the experiments in the main text and supplementary information have been repeated independently with similar results.

## Reporting summary

Further information on research design is available in the Nature Portfolio Reporting Summary linked to this article.

# Data availability

The authors declare that all data supporting of results in this study are available within the paper and its Supplementary Information or from the corresponding author upon reasonable request. Source data are provided with this paper.

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

## Acknowledgements

This work was financially supported by the National Key Research and Development Program of China (2018YFA0703003), the National

Natural Science Foundation of China (52125501, 52205317, 31971272, 81972889), the Key Research Project of Shaanxi Province (2021GXLH-Z-028, 2021LLRH-08), the Natural Science Basis Research Plan in Shaanxi Province of China (2022JQ-523), the Program for Innovation Team of Shaanxi Province (2023-CX-TD-17), the High-Level Talent Recruitment Program of Shaanxi Province, the Fundamental Research Funds for the Central Universities, China Postdoctoral Science Foundation. The authors would like to thank Ming Wang for her assistance in conducting AFM nanoindentation.

## Author contributions

M.M designed the study, conducted experiments, analyzed and interpreted the data and wrote the manuscript. X.L.Q. contributed to cell culture, fluorescence imaging and data collection. Y.B.Z. contributes to EHD printing of scaffold, finite-element analysis and mechanical analysis. B.S.G. contributes to the fabrication of PDMS leaf-venation-inspired microchannel substrate, isolation of neonatal rat CMs and electrophysiological monitoring of CMs. C.L. R.Z.L contributed to data process and visualization. X.L. and H.Z. helped in data interpretation and revised the manuscript. J.K.H., D.C.L. supervised the study, revised the manuscript and helped in data interpretation.

## Competing interests

The authors declare no competing interests.
