## [Peer Review File · Nature Communications]

REVIEWER COMMENTS

Reviewer #1 (Remarks to the Author):

In this manuscript, the authors describe a leaf-venation-directed (LVD) strategy to fabricate cardiac tissue constructs. Using this strategy, they were fabricated constructs using HUVECs which formed interconnected tubular structures within a hierarchical channel distribution. The LVD strategy was also applied for fabrication of cardiac constructs using neonatal rat cardiomyocytes, which also formed interconnected tubular structures with cell alignment; and the fabrication of LVD cardiac constructs using hiPSC-CMs showing cells alignment along the hierarchical channel structure. As control, they used the same types of LVD structures but without the amphiphilic treatment step. The amphiphilic treatment was used to prevent adhesion between the PDMS and cell/hydrogel matrices, where it wasn't used, the cells failed to display the aligned microstructural organization and functional characteristics. Ultimately a multilayer LVD construct was fabricated by applying an elastic PCL scaffold. The methods and findings described represent an improvement over other cardiac constructs. The synchronous beating after 2 days in culture of the LVD-hiPSC-CM is remarkable, as well as the microscale organization shown on day 17. The structural and functional data is evident on day 17; as shown, the LVD construct can be used for drug screening. But, it cannot be concluded that the LVD constructs can be used for long-term culture, that remains to be tested.

Despite the enthusiasm for the technique here described, there are important points of concern. The sample size is missing for all the experiments/figures presented. How many LVD constructs/tissues were included for each of the measurements? It's unclear if the results are based on only one LVD construct/tissue (one from each cell type) with measurements obtained at different areas.

Related to the above, because it is unclear how many tissues were fabricated and analyzed, there is no demonstration of replicability of the results by the investigators. This is a major concern.

Of note, the only experiment where the sample number tested is clear is the mechanical analysis where three samples were tested for each composite configuration (line 751).

Other major concerns include some claims listed in the abstract. Therefore, the abstract and related statements that are included in the introduction and the discussion should be revised.

Abstract:

Line 23: This strategy is not a biomimetic hierarchical vasculature, but rather shows a hierarchical channel distribution with interconnected tubular structures.

Line 25: Robust electrophysiological activity is also questionable. Unclear how many tissues were tested, so replicability and robustness is unclear. Also, calcium transient and field potential experiments were not performed in the LVD-hiPSC-CMs.

The improvements described overall are in comparison to the findings using a similar structure but without amphiphilic treatment, and across time in culture, these are key points that should be stated in the abstract.

Line 28: on the comment of anisotropic mechanical properties of native myocardium, the images show cell alignment, but there is no clear comparison to mechanical properties of the native myocardium. In any case, regarding the result shown in Figure 6D, which tested the viscoelastic properties of the LVD construct, there is no description in the manuscript about the native myocardium tissues that were included in the analysis, are they from rat? Or human? If not, what other comparison was done for this claim?

Line 28-29: Allowing for minimally invasive implantation was not demonstrated. At best, the experiment performed was a proof of concept that the 3D LVD tissue can be rolled into a tube and rushed out of the tubing under running water, maintaining cell viability and recovering its shape.

Several details that should be addressed by the authors are listed below:

Line 54-56. In the context of that sentence, Reference 19 addresses maturation, it doesn't address size or cellular integration.

Line 78: The proposed technology doesn't seem to fit the description for a "green cable". Reference 26 uses the term "green cable" in the context of plants. The authors should define "green cable" in the context of their cardiac constructs, explain why the term "green cable" fits with the description of the proposed system.

Line 95: Please indicate in the subtitle the type of cells that were used in the experiments described in this section; for example: The strategy for engineering LVD tissues using HUVECs.

Line 951: Figure 1. Also, indicate in the Figure title the type of cells used.

Line 101: Indeed, the PDMS substrate has interconnected channels, but they are not exactly biomimetic. More explanation is required to explain what characteristics make it fit the description of being biomimetic.

Line 106 and 107: It is interesting the explanation that the shrinkage of the cell/hydrogel is due to cell-contraction-induced forces. However, there's no evidence that is the actual cause. First, the experiment where this is shown (Figure 1) was done with HUVECs, which are cells that don't contract, unlike the neonatal rat cardiomyocytes or hiPSC-CMs. Have the authors considered other possibilities for this compaction of the cells into a tight assembly? Engineered tissue compaction can also be observed in tissues fabricated with non-contractile cells, such as fibroblasts. Have they tested cells that don't display cell-contraction-induced forces, for example cardiac fibroblast? Do these cells fail to compact into dense cellular structures to prove the point that it is cell-contraction induced forces that is responsible for shrinkage of the cell/hydrogel? Furthermore, the control non-amphiphilic-treated PDMS, with what is expected to be also cells with contraction-induced forces, did not induce shrinkage. Therefore, it is likely that a different reason than contraction-induced forces is responsible for the shrinkage of the cell/hydrogel.

Line 113: this LVD strategy is proposed as a means to engineer a large-scale vascular tissue; what was the size of the LVD construct? Dimensions of width and length and thickness should be provided, for the single LVD constructs and also for the multilayer-LVD construct.

Line 117: gradually compacted probably a better term than gradually contracted, to distinguish from the cell contractility that is displayed by neonatal rat cardiomyocytes and hiPSC-CMs. Throughout the manuscript, when referring to this effect from the HUVECs it would make it more clear if compaction is used, instead of contraction.

Line 958: Figure 1E, at what time point were these SEM images obtained? The compaction of the cell bundles looks far greater in the SEM images than the one shown in 1C-48h.

Fig S2. Line 44, the legend should indicate that this graph is for LVD-HUVEC tissues.

Line 128, 134 and 146. This is describing LVD-HUVEC tissues.

Line 960. Figure 1F-H, should include the marker that was used to stain the cytoskeleton (F-actin).

Line 148. The authors should indicate in the subtitle the types of cells used in the LVD tissues described in this section. For example: The dynamic formation process of LVD-HUVEC tissues.

Line 967. Figure 2 title should also include the type of cells used.

Fig. S4. It would help if it is clarified in the text if the bifurcation region is describing bifurcation of a primary channel, or a bifurcation region within a branch channel.

Line 165. Unclear if micro channel is the same as branch channel. If these are different, a description of the location of each type of channel within the hierarchical architecture of the LVD is needed; a diagram would be helpful.

Line 969 Figure 2B, it should be indicated in the figure legend if this is a main channel or a branch channel.

Line 168 and Figure 2D. The cross-sectional area change is remarkably different than the results presented for the width change (Fig. S2); how was the width measured?

Line 184. The phrase is incomplete: consisting of with randomly-distributed...

Line 203-204. A reference for the stiffness of the native rat myocardium is missing.

Line 988-990. Figure 4C and D, it is missing the reference for the y-axis.

Line 232. Figure 4E, per the description for measurement of calcium spikes these were plotted over 8 seconds (line 224), then, how was the value of the beats per minute calculated?

Line 237. At what time of the LVD fabrication process were the thin platinum wires embedded into the bottom PDMS layer?

Line 991 Figure 4G. The schematic should be improved to better represent the platinum wire embedding; add additional images to the schematic to display if wires are placed into primary and branch channels and how many of them. It is not clear from the diagram what is the distribution of the electrodes.

Line 240 and 241. The 20mV and 30mV values do not match the values displayed in Figure 4I.

Line 243-245. The results shown demonstrate the short-term (up to 4 days) evaluation with the built-in electrodes. It remains to be tested if long-term monitoring is feasible. The sentence should be

rephrased. Why were the built-in electrodes not tested in the LVD-hiPSC-CMs? The LVD –hiPSC-CMs were maintained in culture out to day 17, this offered the opportunity to test if the built-in electrodes could in fact be used for monitoring beyond day 4. This is a question to be addressed in the results for the LVD-hiPSC-CMs, or the discussion.

Line 258. The claim for long-term culture is not demonstrated.

Line 298-299. The authors should include in the discussion a comment on why the genes used for LVD-hiPSC-CM are different than the genes chosen for LVD-neonatal rat (line 216-217).

Figure 5. Did the LVD–hiPSC-CMs also form tubular structures? An image as the on shown in Fig 3C should be provided.

Line 302: The concluding sentence of this section indicates improvement in function. However, improvement in function is not demonstrated in this section (findings in figure 5). Figure 5F shows stable results across time, there is structural improvement, and gene expression as well, but the findings shown in figure 5 are not indicative of function improvement.

Line 314, if these are the LVD fabricated with HUVECs, they should be called LVD-HUVEC instead of LVD vascular tissue. LVD-HUVEC is more accurate name.

Line 314, is the elastic scaffold the EHD printed scaffold?

Line 316 and 319, What is the waiting time for the bonding to occur?

Line 340-343: are the LVD-cardiac tissues shown here from neonatal rat cardiomyocytes or hiPSC-CMs? How many of these multi-layer constructs were fabricated? What was the success rate? Figure 6G, seems to include a LVD-HUVEC layer, followed by LVD-cardiac, followed by LVD-HUVEC, and then again another LVD-CM, this should be rather clearly described. Also for Figure 6G. At what day of culture were each of the LVD structure obtained for the multi-layer construct to be fabricated? How soon after fabrication of the multi-layer structure was the immunofluorescence image obtained? Did the amount of time of incubation with the antibodies change with the addition of more LVD constructs vs the ones shown in the previous figures for a single LVD construct? Does the multilayer LVD construct retain the tubular like structure of the HUVECs and the cardiac cells? Of note it was mentioned above that an image showing if the LVD from hiPSC-CM have tubular structure or not; but if this multilayer-LVD construct included rat-CM it should be shown if it retained the tubular structure or not.

Line 335-337, Figure 6D describes consistency among FEA, experimental and native heart tissue, but there is no description if the native heart tissue is human or rat, and how such native tissue was obtained and processed for the stress-strain tests. This critical information is missing.

Line 353-356, after passing the multilayer LVD-construct through the tube with running water, how was it layered onto the porcine heart? (Fig. 7D) did it require the use of forceps to position it in place? No test was performed for the possibility that the multilayer LVD-construct could slide off, for example in a vertical position of the heart, or with the beating contractions of the heart; in such cases, have the authors considered the need for suturing the multilayer LVD construct onto the heart? Is it suture resistant? Or do they have other approach in mind to keep it in place. There is no mention about integration onto the pig heart. Much still needs to be tested about the possibility of using these LVD constructs for a minimally invasive approach.

Line 1026 Figure 7C, a scale bar would be helpful to draw attention to the size of the construct, at least in the width and length dimensions.

Methods:

Line 541-543. Which culture media was used for the hiPSC-CMs upon thawing and during maintenance?

Line 562: For each of the three cell types the following is missing: concentration of trypsin and time, centrifugation speed and time. And, on which day of differentiation were the hiPSC-CM when trypsinized for LVD tissue formation?

Line 564 and 568: Is the cell-laden fibrinogen suspension added above the substrate? Or is dispensed into the primary channel for distribution into the branch channels? Is the LVD substrate maintained inside a cell culture dish? If yes, which type? How much cell culture medium is required per construct? A photograph of the LVD substrate along with a scale bar would be useful.

Line 565-567: What was the volume of cell-laden suspension added per LVD construct?

Line 652. How many LVD-HUVEC samples were used for the AFM measurements? In other words, how many independent constructs were used in this analysis? Did the areas include primary or branch channels?

Reviewer #2 (Remarks to the Author):

In this study Mao et al., look to utilize the natural branching structures seen in the veins of a leaf to create a branched cardiac tissue worthy of implantation. They utilize a previously reported system of creating a mold from the leaf vein and use both neonatal rat cardiomyocytes and human induced pluripotent stem cell derived cardiomyocytes to create tissues within said mold. Tissues became compact and form tubular structures in a branched larger scaled tissue which allows for anisotropic and vascularized tissue. The created tissue is incredibly well aligned and has impressive contractile and electrical propagation throughout the tissue. Furthermore, the authors looked to implement a minimally invasive strategy to potentially implant the tissue in the future using a mechanical support scaffold. They showed through a proof-of-concept study that the tissue with the mechanical support can be injected via a needle and unrolled through water assistance. Overall, this is an impressive study, and the created tissue is quite commendable. However, there is a question over the vascularization of the tissue. The authors claim the tissue is vascularized but there is limited to no investigation of the claim. Perfusion studies and further structural analysis of the patency of the tissue would help to strengthen these claims of vascularization. Furthermore, the proof-of-concept study was interesting to investigate a minimally invasive way of implanted this tissue, however questions over the tissue survivability and integration post-implant are not addressed and granted this could be from a future larger scaled follow-up animal study.

Major Comments:

1. Further investigation into the claims that the tissue is vascularized is needed. Perfusion of either dyes or microbeads demonstrating patency would help with these claims. Similarly, histological assessment showing the lumens and their sizes at various areas of the tissue would also be beneficial.

Minor Comments:

1. There are a few instances of writing tense being incorrect, please fix.

Response to Reviewers

We wish to thank the reviewers for their detailed and constructive comments, which we feel have improved the clarity and strength of our paper. The reviewers' comments are included in **red italics**, and our responses follow in regular font. In the main text, the changes are indicated in **red**.

Response to Reviewer #1:

In this manuscript, the authors describe a leaf-venation-directed (LVD) strategy to fabricate cardiac tissue constructs. Using this strategy, they were fabricated constructs using HUVECs which formed interconnected tubular structures within a hierarchical channel distribution. The LVD strategy was also applied for fabrication of cardiac constructs using neonatal rat cardiomyocytes, which also formed interconnected tubular structures with cell alignment; and the fabrication of LVD cardiac constructs using hiPSC-CMs showing cells alignment along the hierarchical channel structure. As control, they used the same types of LVD structures but without the amphiphilic treatment step. The amphiphilic treatment was used to prevent adhesion between the PDMS and cell/hydrogel matrices, where it wasn't used, the cells failed to display the aligned microstructural organization and functional characteristics. Ultimately a multi-layer LVD construct was fabricated by applying an elastic PCL scaffold. The methods and findings described represent an improvement over other cardiac constructs. The synchronous beating after 2 days in culture of the LVD-hiPSC-CM is remarkable, as well as the microscale organization shown on day 17. The structural and functional data is evident on day 17; as shown, the LVD construct can be used for drug screening. But, it cannot be concluded that the LVD constructs can be used for long-term culture, that remains to be tested.

We appreciate the reviewer's time for his/her careful reading of the manuscript and the constructive comments that have helped us significantly strengthen this work. We have now addressed the comments, and a detailed point-by-point response to all comments can be found below.

1. Despite the enthusiasm for the technique here described, there are important points of concern. The sample size is missing for all the experiments/figures presented. How many LVD constructs/tissues were included for each of the measurements? It's unclear if the results are based on only one LVD construct/tissue (one from each cell type) with measurements obtained at different areas.

Related to the above, because it is unclear how many tissues were fabricated and analyzed, there is no demonstration of replicability of the results by the investigators. This is a major concern.

Of note, the only experiment where the sample number tested is clear is the mechanical analysis where three samples were tested for each composite configuration (line 751).

We appreciate the reviewer's valuable comments. All the sample sizes for each experimental group/measurement have been stated in the figure legend, and all the column charts have been modified to show the individual data points.

2. Other major concerns include some claims listed in the abstract. Therefore, the abstract and related statements that are included in the introduction and the discussion should be revised.

Abstract:

Line 23: This strategy is not a biomimetic hierarchical vasculature, but rather shows a hierarchical channel distribution with interconnected tubular structures.

We appreciate the reviewer's suggestion. The description '... biomimetic hierarchical vasculatures with interconnected tubular structures' has been replaced by '... interconnected tubular structures with cell alignment along the hierarchical channels' on Page 2, line 22 in the Abstract section.

3. Line 25: Robust electrophysiological activity is also questionable. Unclear how many tissues were tested, so replicability and robustness is unclear. Also, calcium transient and field potential experiments were not performed in the LVD-hiPSC-CMs.

The improvements described overall are in comparison to the findings using a similar structure but without amphiphilic treatment, and across time in culture, these are key points that should be stated in the abstract.

We thank the reviewer for highlighting this point and agree with the reviewer that this description is not accurate. The improvements of electrophysiological activity, including calcium transient and field potential, were only performed in rat CMs compared to the findings using a similar structure but without amphiphilic treatment. To illustrate those points, the relevant texts have been updated as below. The sample sizes for each experimental group/measurement have been stated in the figure legend.

On Page 2, lines 23-27:

'Compared to randomly-distributed cells, the engineered LVD-cardiac tissues from neonatal rat cardiomyocytes manifest advanced maturation and functionality as evidenced by detectable electrophysiological activity, macroscopically synchronous contractions, and upregulated maturation genes. As a demonstration, human induced pluripotent stem cell-derived LVD-cardiac tissues are engineered with evident structural and functional improvement over time.'

4. Line 28: on the comment of anisotropic mechanical properties of native myocardium, the images show cell alignment, but there is no clear comparison to mechanical properties of the native myocardium. In any case, regarding the result shown in Figure 6D, which tested the viscoelastic properties of the LVD construct, there is no description in the manuscript about the native myocardium tissues that were included in the analysis, are they from rat? Or human? If not, what other comparison was done for this claim?

We appreciate the reviewer's professional comments.

- a) The comment on the anisotropic mechanical properties of native myocardium is not accurate here. In Fig. 6B-D, we demonstrated that the mechanical property of the 3D tissue composites could be modulated based on the predesigned structure of the EHD-printed scaffold. Therefore, the description '...constructs resembling the anisotropic mechanical properties of native rat

myocardium...’ has been replaced by ‘...constructs with programmed mechanical properties...’ in the revised manuscript on Page 2, lines 28-29 in the main text.

- b) We are sorry that the crucial reference was missed regarding the mechanical properties of the native myocardium tissues. The stress-strain curve of native heart tissues in Figure 6D is from the adult rat right ventricular myocardium, acquired from previous work (*Nature Materials*, 2008, 7(12):1003-10.). The relevant texts and reference have been updated in the revised manuscript, as below.

On Page 15, lines 339-340:

‘...experimental measurement, and native adult rat heart tissue¹⁰, indicating that...’

5.Line 28-29: Allowing for minimally invasive implantation was not demonstrated. At best, the experiment performed was a proof of concept that the 3D LVD tissue can be rolled into a tube and rushed out of the tubing under running water, maintaining cell viability and recovering its shape.

We thank the reviewer for pointing out this inaccurate description. We have corrected it in the revised manuscript. The description ‘...and allowing for minimally invasive implantation’ has been replaced by ‘...which can be delivered through tubing without affecting cell viability’ on Page 2, lines 29-30 in the main text.

6.Several details that should be addressed by the authors are listed below.

Line 54-56. In the context of that sentence, Reference 19 addresses maturation, it doesn’t address size or cellular integration.

We thank the reviewer for pointing out this. The description regarding Reference 19 has been revised according to the reviewer’s kind suggestion. The relevant texts have been updated as below.

On Page 3, lines 56-57:

‘Furthermore, cell alignment induced by anisotropic fibrous scaffolds alone has a limited effect on improving cardiomyocyte maturation¹⁹’

7.Line 78: The proposed technology doesn’t seem to fit the description for a “green cable”. Reference 26 uses the term “green cable” in the context of plants. The authors should define “green cable” in the context of their cardiac constructs, explain why the term “green cable” fits with the description of the proposed system.

We thank the reviewer for pointing out this inappropriate use. The term “green cable” here would make the description more complex. The description ‘...can be considered as a ‘green cable’ allowing the transmission of action potentials...’ has been replaced by ‘...can work as a critical highway for the transmission of action potentials...’ on Page 4, lines 78-79 in the main text.

8.Line 95: Please indicate in the subtitle the type of cells that were used in the experiments described in this section; for example: The strategy for engineering LVD tissues using HUVECs.

We have made corrections according to the reviewer’s suggestion in the revised manuscript. The relevant texts have been updated as below.

On Page 5, line 96:

‘The strategy for engineering LVD tissues using HUVECs’

9.Line 951: Figure 1. Also, indicate in the Figure title the type of cells used.

We have made corrections according to the reviewer's suggestion in the revised manuscript. The relevant texts have been updated as below.

On Page 41, lines 974-975:

'LVD strategy of generating large-scale tissues with highly aligned populations and tight intercellular connections using HUVECs'

10.Line 101: Indeed, the PDMS substrate has interconnected channels, but they are not exactly biomimetic. More explanation is required to explain what characteristics make it fit the description of being biomimetic.

We agree with the reviewer that the interconnected channels in the PDMS substrate are not exactly biomimetic. As a nature-selected system, leaf venation enables highly efficient fluid transportation by mimicking mammalian cardiovascular systems' hierarchical and bifurcating structures. However, regarding engineering cardiac tissue, we employed the leaf-venation system to direct the cellular alignment, where the description of being biomimetic was overrated. The term 'leaf-venation-inspired' would be more exact than 'biomimetic' here. We have made the correction in the revised manuscript. The relevant texts have been updated as below.

On Page 5, lines 102-103:

'...polydimethylsiloxane (PDMS) substrates with leaf-venation-inspired microchannels, consisting of a primary channel and multiple branch channels...'

11.Line 106 and 107: It is interesting the explanation that the shrinkage of the cell/hydrogel is due to cell-contraction-induced forces. However, there's no evidence that is the actual cause. First, the experiment where this is shown (Figure 1) was done with HUVECs, which are cells that don't contract, unlike the neonatal rat cardiomyocytes or hiPSC-CMs. Have the authors considered other possibilities for this compaction of the cells into a tight assembly? Engineered tissue compaction can also be observed in tissues fabricated with non-contractile cells, such as fibroblasts. Have they tested cells that don't display cell-contraction-induced forces, for example cardiac fibroblast? Do these cells fail to compact into dense cellular structures to proof the point that it is cell-contraction induced forces that is responsible for shrinkage of the cell/hydrogel? Furthermore, the control non-amphiphilic-treated PDMS, with what is expected to be also cells with contraction-induced forces, did not induce shrinkage. Therefore, it is likely that a different reason than contraction-induced forces is responsible for the shrinkage of the cell/hydrogel.

We sincerely thank the reviewer for pointing it out. We are sorry for the improper use of the term 'contraction' here, which tends to be linked with the cell contractility of neonatal rat cardiomyocytes and hiPSC-CMs. It should be the 'cell-spreading-induced forces' that cause the shrinkage of the cell/hydrogel.

We have employed the LVD strategy to culture various non-contractile cells including HUVECs (Human umbilical vein endothelial cells), C2C12 (skeletal myocytes), NHDF(normal human dermal fibroblast). All of them were found to compact into highly aligned and densely packed tissues confined in the PDMS substrate, as shown in the figure below.

Fig. 1R Engineering of LVD-HUVECs, LVD-C2C12, and LVD-NHDF tissues.

Previous studies indicated that the cells cultured in biologically-derived fibrous matrices could spread and exert forces on the surrounding ECM (*Nature materials*, 2015, 14(12): 1262-1268). Owing to the nonlinear-elastic nature of fibrous ECM (e.g., fibrin hydrogel), the cell-spreading-induced forces could lead to the realignment of fibrin hydrogel fibers between cells, which triggered cellular morphological changes (*Nature*, 2005, 435(7039): 191-194; *PNAS*, 2016, 113(49): 14043-14048). In our study, the cells initially distributed in the fibrin hydrogel began to spread with the increase of culture time. They produced forces that induced the cells and hydrogel fibers to gradually gather towards the central region of channels, contributing to the formation of densely-populated and highly-aligned cellular bundles.

We have modified the description and added crucial references in the revised manuscript. The relevant texts have been updated as below.

On Page 5, lines 107-110:

‘The uniformly-encapsulated cells begin to spread and exert traction forces on the surrounding matrix³⁰, initiating LVD morphological evolution process (Fig. 1Aiv,Av). Subsequently, the cell-spreading-induced forces can induce the shrinkage of the cell/hydrogel^{31,32}, which...’

12.Line 113: this LVD strategy is proposed as a means to engineer a large-scale vascular tissue; what was the size of the LVD construct? Dimensions of width and length and thickness should be provided, for the single LVD constructs and also for the multilayer-LVD construct.

We have made corrections according to the reviewer’s suggestion in the revised manuscript. To illustrate the dimensions, the relevant texts have been updated as below.

On Page 6, lines 116-118:

‘The overall size of the generated oval leaf-like LVD tissue is $\sim 760 \text{ mm}^2$, with a length of $\sim 45 \text{ mm}$, a maximum width of 24 mm , and the cellular bundles are $\sim 20\text{-}50 \text{ }\mu\text{m}$ in diameter.’

On Page 15, lines 347-348:

‘...to generate a 4-layer LVD-cardiac tissue construct with a thickness of $\sim 500 \text{ }\mu\text{m}$ (Fig. 6F,G).’

13.Line 117: gradually compacted probably a better term than gradually contracted, to distinguish

from the cell contractility that is displayed by neonatal rat cardiomyocytes and hiPSC-CMs. Throughout the manuscript, when referring to this effect from the HUVECs it would make it more clear if compaction is used, instead of contraction.

We thank the reviewer for this professional comment. To distinguish from the contractility of neonatal rat cardiomyocytes and hiPSC-CMs, ‘compact’ is more exact than ‘contract’ here. We have made corrections according to the reviewer’s suggestion in the revised manuscript. The relevant texts have been updated as below.

On Page 6, line 120:

‘...and gradually compacted in the ...’

14.Line 958: Figure 1E, at what time point were these SEM images obtained? The compaction of the cell bundles looks far greater in the SEM images than the one shown in 1C-48h.

The sample for SEM images was obtained after 3 days of culture in vitro. The reason why the compaction of the cell bundles looks far greater in the SEM images than the one shown in 1C-48h is:1) Fig. 1C-48h shows the cell bundles in the primary channel, while the SEM images in Fig. 1E shows that in branch channels. It has been demonstrated that the average width of the cell bundles in the branch channels was ~2.5-fold thinner than that in the primary channels. 2) The cell/hydrogel constructs suffered further shrinkage during the drying process for preparing SEM samples, which made them thinner.

To illustrate those, the relevant texts have been updated as below.

On Page 42, line 980:

‘(C)...during 48 hours of culture **within primary channels**’

On Page 42, lines 982-983:

‘(E) Representative SEM images of LVD cellular bundles (green) within **branch** channels (purple) cultured for **3 days**.’

15.Fig S2. Line 44, the legend should indicate that this graph is for LVD-HUVEC tissues.

We have made corrections according to the reviewer’s suggestion in the revised manuscript. The relevant texts have been updated as below.

Fig S2, line 39:

‘The width change of the LVD-HUVEC tissues in the primary and branch channel.’

16.Line 128, 134 and 146. This is describing LVD-HUVEC tissues.

We have made corrections according to the reviewer’s suggestion in the revised manuscript.

17.Line 960. Figure 1F-H, should include the marker that was used to stain the cytoskeleton (F-actin).

We have made corrections according to the reviewer’s suggestion in the revised manuscript. The relevant texts have been updated as below.

On Page 42, lines 983-984:

‘The stained F-actin in LVD (F) and control tissues (G)...’

18.Line 148. The authors should indicate in the subtitle the types of cells used in the LVD tissues described in this section. For example: The dynamic formation process of LVD-HUVEC tissues.

We have made corrections according to the reviewer’s suggestion in the revised manuscript. The relevant texts have been updated as below.

On Page 7, line 150:

‘The dynamic formation process of LVD-HUVEC tissues’

19.Line 967. Figure 2 title should also include the type of cells used.

We have made corrections according to the reviewer’s suggestion in the revised manuscript. The relevant texts have been updated as below.

On Page 43, line 991:

‘...the formation process of LVD-HUVEC tissues...’

20.Fig. S4. It would help if it is clarified in the text if the bifurcation region is describing bifurcation of a primary channel, or a bifurcation region within a branch channel.

We have made corrections according to the reviewer’s suggestion in the revised manuscript. The relevant texts have been updated as below.

Fig. S4, lines 49:

‘...in primary channel and bifurcation of branch channels.’

21.Line 165. Unclear if micro channel is the same as branch channel. If these are different, a description of the location of each type of channel within the hierarchical architecture of the LVD is needed; a diagram would be helpful.

In this manuscript, the microchannels are defined as the entire leaf-venation channels, featured with the hierarchical architectures. Therefore, the microchannels include a primary channel and multiple branch channels. We have indicated the location of primary and branch channels in Figure Ai, and revised the relevant description at the beginning of the Results section, as below.

Fig. 1Ai

On Page 5, lines 101-104:

‘As described in our previous works, polydimethylsiloxane (PDMS) substrates with leaf-venation-inspired microchannels, consisting of a primary channel and multiple branch channels, can be produced from the skeleton of leaf-venation networks (Fig. 1Ai)²⁹.’

22.Line 969 Figure 2B, it should be indicated in the figure legend if this is a main channel or a branch channel.

We have made corrections according to the reviewer’s suggestion in the revised manuscript. The relevant texts have been updated as below.

On Page 43, lines 993-994:

‘(B) The sequential steps of LVD self-assembly of HUVECs (magenta)/fibrin hydrogel (cyan) in **branch channel**.’

23.Line 168 and Figure 2D. The cross-sectional area change is remarkably different than the results presented for the width change (Fig. S2); how was the width measured?

We thank the reviewer for pointing out this.

Width measurement: The bright-field images of HUVEC/fibrin hydrogel during the compaction and tissue remodeling process were imported into ImageJ to measure their width.

As shown in Fig. 2B below, the cell/fibrin hydrogel shrinks in both horizontal (y-direction) and vertical (z-direction) dimensions. The width change measured from the bright field image can only indicate the compaction ratio in the horizontal dimension. In contrast, the cross-sectional area measured by confocal files shows the compaction ratio of the cell bundle in both horizontal and vertical dimensions. Therefore, the cross-sectional area change is remarkably different than the results presented for the width change.

The width measurement has been updated in the Method section, as below.

On Page 31, lines 724-726:

‘Specifically, the bright-field images of HUVEC/fibrin hydrogel during the compaction and tissue remodeling process were imported into ImageJ to measure their width.’

Fig. 2B Cross-section of LVD-HUVEC tissue during self-assembly

24.Line 184. The phrase is incomplete: consisting of with randomly-distributed...

We thank the reviewer for pointing out this mistake. The description ‘...consisting of with randomly-distributed fibers...’ has been replaced by ‘...consisting of randomly-distributed fibers...’ on Page 7, lines 186-187 in the main text.

25.Line 203-204. A reference for the stiffness of the native rat myocardium is missing.

We thank the reviewer for the kind reminder, and the reference has been added in the revised manuscript. The relevant texts have been updated as below.

On Page 10, line 206:

‘...which was close to that of the rat adult myocardium (~22-51 kPa)³⁶.’

26.Line 988-990. Figure 4C and D, it is missing the reference for the y-axis.

We thank the reviewer for the kind reminder. The reference for the y-axis in Fig. 4C and 4D have been added in the revised manuscript, as below.

Figure 4C and D

The related description has been added in the Methods as below.

On Page 30, lines 697-698:

‘The measured fluorescence intensity in arbitrary units was plotted against time.’

27.Line 232. Figure 4E, per the description for measurement of calcium spikes these were plotted over 8 seconds (line 224), then, how was the value of the beats per minute calculated?

We thank the reviewer for pointing it out. The spontaneous beating rate of LVD cardiac tissues was determined optically by counting the number of beats per minute in the bright-field mode of an inverted light microscope. To illustrate this, the related description has been updated in the Methods and Figure legend sections as below.

On Page 30, lines 699-700 (Methods):

‘The spontaneous beating rate of LVD and control tissues was determined optically by counting the number of beats per minute in the bright-field mode of an inverted light microscope.’

On Page 47, lines 1021-1023 (Figure legend):

‘(E) The quantification of time to peak of calcium transient correspondingly. n = 3 independent samples. (F) The quantification of beating rate in spontaneously beating LVD and control tissues. n = 3 independent samples.’

28.Line 237. At what time of the LVD fabrication process were the thin platinum wires embedded into the bottom PDMS layer?

The embedding of thin platinum wires: The thin platinum wire was first inserted into the primary channel of the as-prepared PDMS substrate by hand, assisted by a hollow needle. Thus, a short end of the wire exposed in the primary channel works as microelectrodes, while the rest under the PDMS substrate work as the interconnect. Then, the PDMS elastomer was coated onto the bottom of the PDMS substrate to seal the platinum interconnects. After the curing of the coated PDMS elastomer, the embedding of thin platinum wires into the PDMS substrate was completed.

To illustrate this, we simplified the description in the Result section and provided more detailed information in the Methods section. The related description has been updated as below.

on Page 11, lines 236-240 (Results)

‘To demonstrate it, we integrated two thin platinum wires as conductive microelectrodes into the primary channel of LVD system for *in-situ* biologically relevant electrophysiological monitoring of CMs (Fig. 4Gi). During the formation of LVD-cardiac tissues, the microelectrodes could be enclosed by the densely packed cellular bundle (Fig. 4Gii).’

on Page 26, lines 616-628 (Methods)

‘A custom-built microelectrode-integrated device was produced for culturing LVD-cardiac tissue and their *in-situ* biologically relevant electrophysiological assessment. Specifically, the thin platinum wires with an average diameter of 30 μm were placed in a hollow needle and inserted into the primary channel of the as-prepared PDMS leaf-venation-inspired microchannel substrate by hand. The hollow needle was then removed, leaving the thin platinum wire passing through the PDMS layer. The end of the integrated thin platinum wires with a length of $\sim 1\text{mm}$ was exposed to the cells in primary channels, working as microelectrodes to acquire their electrical activity. After that, the PDMS elastomer was coated onto the bottom of the PDMS leaf-venation-inspired microchannel substrate to seal the platinum interconnects. The conductive silver paste was pipetted at the contact pad to connect the platinum to the electrophysiological measurement setup. The obtained microelectrode-

integrated PDMS substrates could be autoclaved and stored for subsequent cell culture. Each PDMS substrate was integrated with two microelectrodes in the primary channel in this work.'

29.Line 991 Figure 4G. The schematic should be improved to better represent the platinum wire embedding; add additional images to the schematic to display if wires are placed into primary and branch channels and how many of them. It is not clear from the diagram what is the distribution of the electrodes.

We thank the reviewer for the wise suggestion. The schematic in Figure 4G has been revised to represent the platinum wire embedding and display the position of the electrodes in the PDMS substrate. The new schematics are shown in the following Figures, which can be found in Figure 4G.

To clarify the number and distribution of the microelectrodes, the relevant texts and figures have been updated in the Result section as below.

On Page 11, lines 236-240:

'To demonstrate it, we integrated two thin platinum wires as conductive microelectrodes into the primary channel of LVD system for *in-situ* biologically relevant electrophysiological monitoring of CMs (Fig. 4Gi). During the formation of LVD-cardiac tissues, the microelectrodes could be enclosed by the densely packed cellular bundle (Fig. 4Gii).'

Fig. 4G

In our work, we just inserted two microelectrodes in the primary channel of each PDMS substrate, as a demonstration to verify the feasibility of *in-situ* electrophysiological assessment of LVD cardiac tissues in our system. In our experience, the platinum wires could successfully be inserted into the PDMS substrate's primary channel by hand, which was more than 500 μm in width. However, it is challenging to insert the platinum wires exactly into the branch channels, whose widths were mostly less than 150 μm . Some specialized positioning devices should be used to assist the operation. Moreover, the diameter of the cellular bundles in the branch channels is mostly ~ 50 μm . The effect of the 30- μm - diameter microelectrodes on them still needs further investigation. We have illustrated this concern in the Discussion section, as below.

On Page 19, lines 433-441 (Discussion):

‘One concern is the limited distribution of microelectrodes in the LVD system. In this work, we just inserted two microelectrodes in the primary channel of each PDMS substrate as a demonstration. In our experience, the platinum wires could be successfully inserted into the PDMS substrate’s primary channel by hand. However, it is challenging to insert the platinum wires exactly into the branch channels, whose widths were mostly less than 150 μm . Some specialized positioning devices should be developed to assist the operation. Moreover, the effect of the 30- μm -diameter microelectrodes on the growth of the cellular bundles in the branch channels still needs further investigation since their final diameter is mostly less than 50 μm .’

30.Line 240 and 241. The 20mV and 30mV values do not match the values displayed in Figure 4I.

We thank the reviewer for pointing out this mistake. The description ‘...LVD rat cardiac tissues in 20 mV spaced with a frequency of ~ 0.5 Hz, whose amplitude increased to 30 mV with...’ has been replaced by ‘...LVD-rat-cardiac tissues in ~ 40 mV spaced with a frequency of ~ 1 Hz, whose amplitude increased to ~ 60 mV with...’ on Page 11, lines 240-242 in the main text.

31.Line 243-245. The results shown demonstrate the short-term (up to 4 days) evaluation with the built-in electrodes. It remains to be tested if long-term monitoring is feasible. The sentence should be rephrased. Why were the built-in electrodes not tested in the LVD-hiPSC-CMs? The LVD – hiPSC-CMs were maintained in culture out to day 17, this offered the opportunity to test if the built-in electrodes could in fact be used for monitoring beyond day 4. This is a question to be addressed in the results for the LVD-hiPSC-CMs, or the discussion.

We thank the reviewer for the professional comments. The term ‘long-term’ is not suitable here. Therefore, the description ‘...enables long-term noninvasive monitoring of the natural maturation and other rhythmic cellular phenomena of the cardiac tissues, which might be used for long-term online cardiotoxicity testing ...’ has been replaced by ‘...enables noninvasive monitoring of the cardiac tissues’ natural maturation and other rhythmic cellular phenomena, showing the potential in online cardiotoxicity testing ...’ on Page 11, lines 244-245 in the main text.

In this work, we employed rat CMs to introduce the LVD strategy of engineering cardiac tissues, demonstrating its advances in improving tissue maturation (e.g. elongated and aligned architecture, synchronous contractions, electrophysiological activity, and several crucial maturation genes), assembly of 3D tissue composites with the programmed mechanical property, as well as delivery through the tubing. hiPSC-CMs were employed to preliminarily demonstrate the feasibility of engineering functional human cardiac tissue using the LVD strategy. Therefore, not all the characteristics of LVD cardiac tissues of hiPSC-CMs were investigated, such as the in-situ electrophysiological monitoring using the built-in electrodes. However, it would be valuable to conduct an in-depth study of culturing hiPSC-CMs in the LVD system with built-in micropillar electrodes in the future. Our system might provide a useful tool for long-term noninvasive monitoring of the LVD human cardiac tissues’ maturation and other rhythmic cellular phenomena from hiPSC-CMs. We have introduced the value of culturing hiPSC-CMs in the microelectrode-

integrated LVD system in the Discussion section, as below.

On Page 19, lines 425-433 (Discussion):

‘We have demonstrated the feasibility of the LVD system with built-in conductive microelectrodes to provide electrical stimulation for the cultured rat CMs and further monitor their extracellular field potentials in situ. It is of great significance to conduct an in-depth study of culturing hiPSC-CMs in the microelectrode-integrated LVD system in the future, considering their potential as an attractive cell source for drug development and regenerative therapy applications. Besides the environmental cues, there are other methods to improve the maturation of hiPSC-CMs, such as electrical/mechanical stimulation and long-term culture with evolving media supplementation^{51, 52}. As a noninvasive approach, our microelectrode-integrated LVD system would have the superiority of culturing matured hiPSC-CMs-based cardiac tissues with functionality.’

32.Line 258. The claim for long-term culture is not demonstrated.

We agreed with the reviewer that the ‘long-term culture’ is overrated here. The description ‘...as well as long-term culture and maturation of cardiac cells.’ has been replaced by ‘...as well as *in-situ* monitoring of electrophysiological activities of the cardiac tissues.’ on Page 12, lines 258-259 in the main text.

33.Line 298-299. The authors should include in the discussion a comment on why the genes used for LVD-hiPSC-CM are different than the genes chosen for LVD-neonatal rat (line 216-217).

We thank the reviewer for this comment. We chose the genes based on previous studies: **1)** The genes, including GJA1, KCNJ2, PDK4, CPT1B, and PPARGC1A, have been successfully used for qRT-PCR to characterize the maturation of neonatal rat CM in the following studies: *Nature communications*, 2011, 2(1): 300; *Nature biotechnology*, 2013, 31(1): 54-62; *Environmental health perspectives*, 2012, 120(9): 1243-1251; *Current Research in Physiology*, 2022, 5: 55-62. **2)** The genes, including S100A1, PLN, SCN5A, COX6A2, CKM, and PDK4, have been successfully used for qRT-PCR to characterize the maturation of hiPSC-CM in the following studies: *Nature communications*, 2017, 8(1): 1825; *Frontiers in Cell and Developmental Biology*, 2022, 10; *Physiological genomics*, 2012, 44(4): 245-258.)

The qRT-PCR results in our study have successfully demonstrated the maturation of LVD rat tissues in comparison to the control group and revealed the maturation of LVD human tissues across time in culture. However, as the reviewer’s comment indicates, we think it would be better to choose the same genes for the rat CMs and hiPSC-CMs in one study. For this, each gene should be carefully checked to ensure they are highly expressed in both rats and humans.

34.Figure 5. Did the LVD–hiPSC-CMs also form tubular structures? An image as the on shown in Fig 3C should be provided.

Thanks for the kind suggestion. The cross-section of the confocal file indicated that hiPSC-CMs

also formed tubular structures, as shown in Fig S. below.

The relevant texts and figures have been updated as below.

On Page 13, lines 289-290:

‘The cross-section of the confocal file indicated that the hiPSC-CMs also form a tubular structure (Fig. S6).’

Page 7 in Supplementary Information:

Fig. S6 Confocal files showing the tubular structures of LVD-cardiac tissue from hiPSC-CMs.

35.Line 302: The concluding sentence of this section indicates improvement in function. However, improvement in function is not demonstrated in this section (findings in figure 5). Figure 5F shows stable results across time, there is structural improvement, and gene expression as well, but the findings shown in figure 5 are not indicative of function improvement.

We agree with the reviewer that ‘improved electrical and metabolic function’ has not been demonstrated by the findings in Figure 5. The relevant descriptions have been modified in the revised manuscript and updated as below.

On Page 13, lines 301-303:

‘5 of the 6 genes progressively increased at day 17 in LVD tissues compared to day 2, suggesting enhanced LVD hiPSC-derived cardiac tissue maturation over time in culture (Fig. 5K).’

36.Line 314, if these are the LVD fabricated with HUVECs, they should be called LVD-HUVEC instead of LVD vascular tissue. LVD-HUVEC is more accurate name.

We have made corrections according to the reviewer’s suggestion in the revised manuscript. The relevant texts have been updated as below.

On Page 14, lines 315-316:

‘...the well-formed LVD-HUVEC tissue in the PDMS substrate...’

37.Line 314, is the elastic scaffold the EHD printed scaffold?

The elastic scaffold is an EHD-printed scaffold. To illustrate this, the relevant texts have been updated as below.

On Page 14, line 315:

‘...an EHD-printed elastic scaffold was placed...’

38.Line 316 and 319, What is the waiting time for the bonding to occur?

The waiting time for the bonding is 5 minutes. The relevant texts have been updated as below.

On Page 14, lines 317:

‘...and polymerized for 5 minutes at 37°C to encapsulate the scaffold and tissue...’

On Page 14, lines 320-322:

‘...with 100 μ L bovine fibrin precursor solution added between them. After polymerizing for 5 minutes at 37°C, the fibrin hydrogel could work as glue to bond the LVD tissues together (Fig. 6Aiv).’

39.Line 340-343: are the LVD-cardiac tissues shown here from neonatal rat cardiomyocytes or hiPSC-CMs? How many of these multi-layer constructs were fabricated? What was the success rate? Figure 6G, seems to include a LVD-HUVEC layer, followed by LVD-cardiac, followed by LVD-HUVEC, and then again another LVD-CM, this should be rather clearly described. Also for Figure 6G. At what day of culture were each of the LVD structure obtained for the multi-layer construct to be fabricated? How soon after fabrication of the multi-layer structure was the immunofluorescence image obtained? Did the amount of time of incubation with the antibodies change with the addition of more LVD constructs vs the ones shown in the previous figures for a single LVD construct? Does the multi-layer LVD construct retain the tubular like structure of the HUVECs and the cardiac cells? Of note it was mentioned above that an image showing if the LVD from hiPSC-CM have tubular structure or not; but if this multilayer-LVD construct included rat-CM it should be shown if it retained the tubular structure or not.

We thank the reviewer for this insightful comment.

a. Rat CMs or hiPSC-CMs

The LVD-cardiac tissues shown in Figure 6E-G are from neonatal rat cardiomyocytes. The relevant texts have been updated in the Result and Figure legend, as below.

On Page 15, line 343:

‘Fig. 6E shows the transferred monolayer LVD-cardiac tissue from rat CMs after 4 days of culture...’

b. Number of multi-layer constructs

We successfully fabricated more than 12 multi-layer constructs after the operation process was fixed. In our experience, the success rate is 100% with the careful operation. We think it was owing

to the introduction of an EHD-printed scaffold which reinforced the LVD tissues/fibrin hydrogel construct and made the manipulation simple

c. Culture days of monolayer LVD tissue and staining of multi-layer construct

We have now introduced Fig. 6G with more details. Regarding Fig. 6G, we employed CellTracker™, a fluorescent dye for live cell tracking, to stain LVD tissues, so no immunofluorescence staining was involved here. Specifically, the 4-day-old monolayer LVD-cardiac and 3-day-old LVD-HUVEC tissues were pre-labeled by red and green CellTracker™ respectively, and then superimposed with each other to form a 4-layer tissue construct. To illustrate this, the relevant texts have been updated in the Result section and Figure caption, as below.

On Page 15, lines 345-348 (Result):

‘As a demonstration, 4-day-old LVD-rat-cardiac and 3-day-old LVD-HUVEC tissues were stained by red and green cell trackers respectively, and then superimposed with each other to generate a 4-layer LVD-cardiac tissue construct with a thickness of ~500 μm (Fig. 6F,G).’

On Page 52, lines 1057-1059 (Figure legend):

‘(F and G) The macroscopic (F) and confocal (G) images of the assembled 4-layer LVD-cardiac construct, consisting of red LVD-rat-cardiac and green LVD-HUVEC tissues superimposed with each other.’

d. Tubular-like structure in multi-layer LVD constructs

We agree with the reviewer that the LVD-HUVECs and the rat CMs in the multi-layer LVD tissue construct should retain their tubular-like structure. The engineered LVD tissues are tubular structures with confluent cells encircling the heavily-compacted fibrin fiber, not the weak lumen structures. Therefore, the heavily-compacted fibrin fibers can provide cell-binding sites for firm cell attachment and high mechanical strength to protect the tubular structure while assembling the multi-layer LVD tissues. It can be found that the cellular bundles are well maintained with clear borders and interconnected structures in the assembled multi-layer constructs (Fig. 6F), even in the constructs which have been rolled and released through tubing (Fig. 7E and 7H). This implies that the tubular-like structures of cellular bundles in the assembled 3D construct were not damaged. Therefore, we did not intentionally characterize the tubular-like structures in the 3D construct.

Fig.6G aims to show the distribution of LVD-rat-cardiac and LVD-HUVEC tissues in the assembled 3D construct, which were prelabelled by the red and green Celltracker respectively before assembly. However, probably because the Celltracker dyes tend to leak outside of the cell and stain neighboring cells (*ACS Chem. Biol.* 2020, 15, 6, 1613–1620), it is difficult to distinguish the tubular-like structure from the cross-sectional confocal images of the 3D construct with a low magnification (10x objective), as shown in Fig. 6G.

Fig. 6F and Fig. 6G

Fig. 7E and Fig. 7H

40. Line 335-337, Figure 6D describes consistency among FEA, experimental and native heart tissue, but there is no description if the native heart tissue is human or rat, and how such native tissue was obtained and processed for the stress-strain tests. This critical information is missing.

We thank the reviewer for pointing it out. The stress-strain curve of native heart tissues is from adult rat right ventricular myocardium, acquired from previous work (*Nature Materials*. 2008, 7(12):1003-10.). The reference has been added in the revised manuscript, as below.

On Page 15, lines 339-340:

‘...experimental measurement, and native adult rat heart tissue¹⁰, indicating that...’

41. Line 353-356, after passing the multi-layer LVD-construct through the tube with running water; how was it layered onto the porcine heart? (Fig. 7D) did it require the use of forceps to position it in place? No test was performed for the possibility that the multi-layer LVD-construct could slide off, for example in a vertical position of the heart, or with the beating contractions of the heart; in such cases, have the authors considered the need for suturing the multi-layer LVD construct onto the heart? Is it suture resistant? Or do they have other approach in mind to keep it in place. There is no mention about integration onto the pig heart. Much still needs to be tested about the possibility of using these LVD constructs for a minimally invasive approach.

We thank the reviewer for the professional comment.

a. How to layer LVD-construct onto the porcine heart

Passing the multi-layer LVD construct through the tube with running water requires a tool such as

forceps to position it on the heart. Owing to the mechanical reinforcement of the EHD-printed PCL scaffold, the construct could easily cover the *ex vivo* heart with a conformable and tight attachment without destroying their structures. The relevant texts have been updated as below.

On Page 16, lines 359-361:

‘With placement by forceps, the injected cardiac construct was capable of covering a clinically-relevant size of the porcine heart...’

b. Potential fixation methods of LVD construct on heart

Fig. 7 aims to investigate the feasibility of delivery of 3D constructs through the tubing, under the mechanical aid of an EHD-printed scaffold. Therefore, in this study, we did not perform the test to affix the construct on beating hearts. Regarding this issue, suturing the LVD construct onto the ventricle might be a choice when doing open-chest surgery. However, not all patients would qualify for this surgery, and a minimally invasive approach is usually preferred. In this case, passing the LVD construct through a tube and affixing it using a fibrin glue might be a potential approach, as a minimally invasive approach.

We have conducted a preliminary *ex vivo* experiment to attach the LVD construct onto the porcine heart using fibrin glue. The process is shown in the new Supplementary Movie S6, and the following representative screen captures (Fig. S12). It was demonstrated that the injected LVD construct was conformably and firmly fixed on the heart, no matter that they suffered random rotation and mechanical stretch. Previous work also proved that fibrin glue could fix the cardiac patch to the beating heart during the *in vivo* patch delivery using a similar approach (*Nature materials*, 2017, 16(10): 1038-1046.).

In the future, more works still need to be done systematically to demonstrate the entire process of integrating these LVD constructs to a porcine heart as a minimally invasive approach. In the revised manuscript, we discussed this topic and the potential fibrin-glue-based method for affixing the multi-layer LVD constructs to the heart.

Fig. S12 The representative screen captures of injection and fibrin-based fixation of the LVD construct to the *ex vivo* porcine heart.

The relevant texts have been updated as below to introduce the fibrin-based fixation method.

On Page 16, lines 365-369:

‘Previously, fibrin hydrogel has been used as glue to secure the cardiac patch on the heart in vivo⁴³. We then conducted a preliminary study to attach the 3D LVD tissue constructs onto the ex vivo porcine heart using fibrin glue. The result demonstrated firm fixation between them, even when they suffered from rotation and mechanical stretch (Movie S6 and Fig. S12).’

42.Line 1026 Figure 7C, a scale bar would be helpful to draw attention to the size of the construct, at least in the width and length dimensions.

We thank the reviewer for the kind suggestion, and the scale bar has been added in the new Figure 7C, as shown below.

Figure 7C

43.Methods:

Line 541-543. Which culture media was used for the hiPSC-CMs upon thawing and during maintenance?

We used the commercial kit for culturing hiPSC-CMs, including cell thawing medium, coating solution, cardiac maintenance medium. The relevant texts have been updated as below.

On Page 24, lines 556-569 (Methods):

‘hiPSC-CMs (HELP4111, NovoCell™) were purchased from Help Therapeutics (Nanjing, China) and cultured according to manufacturer’s instructions. Briefly, a six-well cell culture plate was added with coating solution (F00201, Help Therapeutics) of 1ml in a single well and placed in a 37°C incubator. After 1 hour, the coating solution was removed and changed to 1ml cell thawing medium (F00901, Help Therapeutics) preheated at room temperature for 30 min. According to manufacturer, NovoCell™ hiPSC-CMs were produced by differentiating hiPSCs for 18 days, and consisted mainly of ventricular cells with autonomous electrophysiological activity. Cryovials of hiPSC-CMs were quickly thawed in a 37°C water bath, transferred into a 15 ml centrifuge tube, and added with a 10 ml preheated cell thawing medium. The hiPSC-CMs suspension was then centrifuged at 300×g for 5 min, re-suspended with cell thawing medium, added to the pre-coated six-well plate with 2×10⁶ cells in a single well, and cultured in cardiac maintenance medium (F00301, Help Therapeutics). All culture

media contained 1% penicillin-streptomycin. After 3 days of culture, cells were dissociated for engineering LVD tissue.'

44.Line 562: For each of the three cell types the following is missing: concentration of trypsin and time, centrifugation speed and time. And, on which day of differentiation were the hiPSC-CM when trypsinized for LVD tissue formation?

We thank the reviewer for pointing it out. The information regarding the concentration of trypsin and time, centrifugation speed, and time for each cell type has been added. The relevant texts have been updated as below.

On Page 25, lines 588-592 (Methods):

'To prepare cells for seeding, HUVECs were trypsinized with 0.05% trypsin-EDTA for 1-2 min and centrifuged at 350×g for 5 min; rat CMs were trypsinized with 0.25% trypsin-EDTA for 1-2min and centrifuged at 220×g for 5 min; hiPSC-CMs were trypsinized with specified trypsin (F00101, Help Therapeutics) for 3-5 min and centrifuged at 300×g for 5min. The obtained cells (HUVECs/rat CMs/hiPSC-CMs) were suspended...'

According to the manufacturer, the hiPSC-CMs were produced by differentiating hiPSCs for 18 days. Then, after thawing and 3 days of culture, they were dissociated for engineering LVD tissue. We have specified the information regarding hiPSC-CMs culture in the revised manuscript. The relevant texts have been updated as below.

On Page 24, lines 561-569 (Methods):

'According to manufacturer, NovoCell™ hiPSC-CMs were produced by differentiating hiPSCs for 18 days, and consisted mainly of ventricular cells with autonomous electrophysiological activity. ... The hiPSC-CMs suspension was then centrifuged at 300×g for 5 min, re-suspended with cell thawing medium, added to the pre-coated six-well plate with 2×10⁶ cells in a single well, and cultured in cardiac maintenance medium (F00301, Help Therapeutics). All culture media contained 1% penicillin-streptomycin. After 3 days of culture, cells were dissociated for engineering LVD tissue.'

45.Line 564 and 568: Is the cell-laden fibrinogen suspension added above the substrate? Or is dispensed into the primary channel for distribution into the branch channels? Is the LVD substrate maintained inside a cell culture dish? If yes, which type? How much cell culture medium is required per construct? A photograph of the LVD substrate along with a scale bar would be useful.

We thank the reviewer for raising this comment.

a. Adding cell-laden fibrinogen suspension in LVD substrate

The cell-laden fibrinogen suspension was added above the center of the substrate. In the PDMS substrate pre-immersed in 1% (wt/vol) pluronic F-127 solution and washed with PBS solution, most cell-laden fibrinogen suspension can automatically disperse and fill all the microchannels. Some excess cell suspension was scraped off with a cover glass.

b. LVD substrate during cell culture

During cell culture, the PDMS substrate was usually placed inside a cell culture dish with a

diameter of 10 cm. A chamber exists in the PDMS substrate, and a 5-mL culture medium was added to the chamber for culturing the LVD tissues. The photograph of the LVD substrate filled with 5-mL culture medium is shown below, which was also added in Fig. S14.

Fig. S14 PDMS leaf-venation-inspired microchannel substrate added with 5 ml culture medium for cell culture.

To illustrate those, the relevant texts have been updated regarding the seeding and culture of cell-laden fibrinogen suspension in LVD substrate, as below.

On Page 25, lines 595-601:

‘A 120 μ L cell-laden suspension was quickly added over the center of each leaf-venation-inspired microchannel substrate, which was placed in a cell culture dish with a diameter of 10 cm. While most cell-laden suspensions would automatically disperse and fill all the microchannels, some excess cell suspensions were scraped off with a cover glass. After incubation at 37 °C for 15 min to allow fibrinogen gelation, 5 mL culture medium was added into the chamber of the PDMS substrate (Fig. S14) and refreshed every 2 days.’

46.Line 565-567: What was the volume of cell-laden suspension added per LVD construct?

120- μ L cell-laden suspension was added per LVD construct. The relevant texts have been updated as below.

On Page 25, lines 595-597:

‘A 120 μ L cell-laden suspension was quickly added over the center of each leaf-venation-inspired microchannel substrate...’

47.Line 652. How many LVD-HUVEC samples were used for the AFM measurements? In other words, how many independent constructs were used in this analysis? Did the areas include primary or branch channels?

Three independent samples were used for the AFM measurements. Indentation tests were conducted in the areas of both primary and branch channels, and no significant difference was found between them. Therefore, we made no distinction between primary and branch channels during the result analysis. The relevant texts have been updated as below.

On Page 44, line 997:

‘(E) Average Young’s modulus of LVD tissues postseeding and at day 2. n = 3 independent samples.’

On Page 29, lines 688-689:

‘Three force maps were collected in separate areas on each sample and averaged for the sample value with no distinction between primary and branch channels.’

Response to Reviewer #2:

In this study Mao et al., look to utilize the natural branching structures seen in the veins of a leaf to create a branched cardiac tissue worthy of implantation. They utilize a previously reported system of creating a mold from the leaf vein and use both neonatal rat cardiomyocytes and human induced pluripotent stem cell derived cardiomyocytes to create tissues within said mold. Tissues became compact and form tubular structures in a branched larger scaled tissue which allows for anisotropic and vascularized tissue. The created tissue is incredibly well aligned and has impressive contractile and electrical propagation throughout the tissue. Furthermore, the authors looked to implement a minimally invasive strategy to potentially implant the tissue in the future using a mechanical support scaffold. They showed through a proof-of-concept study that the tissue with the mechanical support can be injected via a needle and unrolled through water assistance. Overall, this is an impressive study, and the created tissue is quite commendable. However, there is a question over the vascularization of the tissue. The authors claim the tissue is vascularized but there is limited to no investigation of the claim. Perfusion studies and further structural analysis of the patency of the tissue would help to strengthen these claims of vascularization. Furthermore, the proof-of-concept study was interesting to investigate a minimally invasive way of implanted this tissue, however questions over the tissue survivability and integration post-implant are not addressed and granted this could be from a future larger scaled follow-up animal study.

We thank the reviewer for his or her thorough evaluation of the manuscript and the valuable comments. We agree with the reviewer that a larger-scale follow-up animal study should be conducted in the future to address the issues regarding tissue survivability and integration post-implant systematically. Besides, we have conducted a perfusion experiment and revised the manuscript extensively to address the reviewer's concerns.

Major Comments:

1. Further investigation into the claims that the tissue is vascularized is needed. Perfusion of either dyes or microbeads demonstrating patency would help with these claims. Similarly, histological assessment showing the lumens and their sizes at various areas of the tissue would also be beneficial.

We thank the reviewer for the insightful comments. The claim of vascularized tissues was improper here. We have replaced the term 'vascularized' with 'pre-vascularized' in our revised manuscript.

As the reviewer suggested, we have conducted the perfusion study in our LVD-HUVEC tissue, which was sandwiched by gelatin gel. Specifically, when the culture medium was removed, the well-formed LVD vascular tissue in the PDMS substrate was added with 4 mL of 10 mg/mL gelatin solution (Fig. S15Ai) and placed at 4°C for 5 minutes to form a gel (Fig. S15Aii). Then, this construct was turned over (Fig. S15Aiii), and the PDMS substrate was removed, leaving the LVD vascular tissue within gelatin gel (Fig. S15Aiv). After that, another 4 mL of 5 mg/mL gelatin solution was added to them and placed at 4°C for 5 minutes, resulting in LVD vascular tissue tightly encapsulated in gelatin gel (Fig. S15Av). Finally, an arc-shaped inlet and an outlet were cut out to expose more cellular bundles for perfusing nanobeads solution (Fig. S15Avi). Then, the obtained hydrogels were placed on an inclined plane, and fluorescent nanobeads with a diameter of 80 nm (7-5-1000, Tianjin BaseLine) were pipetted to the inlet of the LVD tissue/gelatin gel

construct (Fig. S15B). Confocal images were taken at different locations inside the construct to investigate whether the nanobeads could be perfused through the LVD-HUVEC networks.

Fig. S15 Schematic illustration of perfusion of nanobeads solution in LVD-HUVEC tissues. (A) Generation of gelatin constructs encapsulating LVD-HUVEC networks for perfusion experiment. (B) The setup for perfusion of fluorescence nanobeads solution through LVD-HUVEC networks.

Fig. S16 shows the representative images of nanobeads distribution in the inlet region and the inner regions. It indicates that only a few nanobeads appeared in the inlet region of the LVD tissue/gelatin gel construct (Fig. S16B). In contrast, no nanobeads were observed in the inner

regions, demonstrating the occlusion of the tubular structures of LVD-HUVEC tissue. This is in accordance with the structural assessment in Fig. 2. It can be found that the interconnected tubular structures of confluent cells in LVD-HUVEC tissue were entirely filled with the heavily-compressed fibrin fibers, which induced the occlusion of the tubular structures.

Fig. S16 Distribution of the perfused blue nanobeads in LVD tissue. (A) Schematic illustration of the LVD tissue/gelatin gel construct with inlet and inner regions. (B) Distribution of blue nanobeads in the inlet region. White arrows indicate some blue nanobeads in the gaps between LVD tissue (identified by dense green fibrin) and gelatin gel, which might be due to the translocation of LVD tissues during the cutting process. (C) No blue nanobeads were found in the inner region.

In a word, LVD-HUVEC tissue was a tubular structure with cells encircling the dense fibrin fibers, not a lumen structure that allows for perfusion *in vitro*. However, these LVD-HUVEC tissues are expected to guide the formation of neo-vasculature with the biomimetic hierarchical architecture and induce ingrowth and anastomoses of host vasculature *in vivo*, which can facilitate the mass transfer. In previous works, a similar tubular structure with specific patterns was generated by seeding cell/matrix mixtures into grooved templates, which would guide their self-assembly (*Tissue Engineering Part C-Methods* 2015, 21, 509-517). It was demonstrated that such self-assembled tubular structures with a prescribed geometry *in vitro* can provide a template that defines the neovascular architecture *in vivo* on implantation and enhance anastomosis and vascular functionality (*PNAS*, 2013, 110(19): 7586-7591). For example, parallel arrays of cords resulted in a similar capillary network, while a single cord with a bifurcation resulted in a perfused branch point. However, the described tubular structures are in simple patterns that tend to shrink during the self-assembly process.

In this study, we attempted to employ the LVD strategy to engineer large-scale, highly interconnected, and hierarchical microvascular tissues that capture the complexity of the native EC arrangement in tubular branching structures. Considering that previous work has demonstrated the ability of the tubular vasculature to template the neovascular architecture *in vivo*, we assume that the LVD-HUVEC tissue can guide the formation of neo-vasculature with the biomimetic

hierarchical architecture and induce ingrowth and anastomoses of host vasculature in vivo which can facilitate the mass transfer. In future work, we will conduct animal experiments to demonstrate the merit of our LVD-HUVEC tissue in enhancing anastomosis and vascular functionality in vivo.

The relevant texts have been updated in the Result, Discussion, and Supplementary information sections, as below.

On Page 8, lines 180-183 (Results):

‘The LVD-HUVEC tissues were then sandwiched by gelatin gel and perfused with nanobeads solution. The result demonstrated the occlusion of the tubular structures, which should be owing to their heavily-compressed fibrin fibers inside (Supplementary result).’

The relevant texts have been introduced in the Discussion section, as below.

On Page 19, lines 442-467 (Discussion):

‘Vascularization of engineered tissues is of the essence for cell survival and function, especially for CMs with a high-metabolic rate. To facilitate an efficient mass transfer, all the CM bundles in the native myocardium are wrapped around by the dense networks of capillaries, whose functionality are relied heavily on their tree-like hierarchical branching structure⁷. Early attempts to create vascularized cardiac tissues through the simple addition of vascular endothelial cells showed that some level of spontaneous-assembly of tubular structures in ECM occurs; however, these resultant vascular components are usually random, uncontrollable, and less efficient for facilitating in-vivo regeneration and anastomosis of perfusable blood vessels^{53, 54}. Although perfusable vascularized structures can be engineered by seeding and pre-organizing endothelial cells along the inner surfaces of the defined microfluidic channels, it remains challenging to fabricate endothelialized channels with biomimicking hierarchical structures at the microvasculature size scale (<30 μm), as well as to integrate them with the functional cardiac tissues^{55, 56}. Another potential strategy for engineering vasculature is seeding cell/matrix mixtures into grooved templates, guiding the self-assembly of cell/matrix mixtures into the tubular structure with specific patterns³⁴. Previous works indicate that such self-assembled tubular structures with a prescribed geometry in vitro can provide a template that defines the neovascular architecture in vivo on implantation and enhance anastomosis and vascular functionality³³. Parallel arrays of cords resulted in a similar capillary network, while a single cord with a bifurcation resulted in a perfused branch point. However, the described tubular structures are in simple patterns that tend to shrink during the self-assembly process. In this study, we attempt to employ the LVD strategy to engineer large-scale, highly interconnected, and hierarchical microvascular tissues that capture the complexity of the native EC arrangement in tubular branching structures. Considering that previous work has demonstrated the ability of the tubular vasculature to template the neovascular architecture in vivo, we assume that the LVD-HUVEC tissue can guide the formation of neo-vasculature with the biomimetic hierarchical architecture and induce ingrowth and anastomoses of host vasculature in vivo which can facilitate the mass transfer.’

Supplementary information, Page 18:

‘Supplementary result

Perfusion of nanobeads solution in LVD-HUVEC tissues

Perfusion of nanobeads solution was conducted to investigate the patency of the LVD-HUVEC tissues. Specifically, when the culture medium was removed, the well-formed LVD vascular tissue in the PDMS substrate was added with 4 mL of 10 mg/mL gelatin solution (Fig. S15Ai) and placed at 4°C for 5 minutes to form a gel (Fig. S15Aii). Then, this construct was turned over (Fig. S15Aiii), and the PDMS substrate was removed, leaving the LVD vascular tissue within gelatin gel (Fig. S15Aiv). After that, another 4 mL of 5 mg/mL gelatin solution was added to them and placed at 4°C for 5 minutes, resulting in LVD vascular tissue tightly encapsulated in gelatin gel (Fig. S15Av). Finally, an arc-shaped inlet and an outlet were cut out to expose more cellular bundles for perfusing nanobeads solution (Fig. S15Avi). Then, the obtained hydrogels were placed on an inclined plane, and fluorescent nanobeads with a diameter of 80 nm (7-5-1000, Tianjin BaseLine) were pipetted to the inlet of the LVD tissue/gelatin gel construct (Fig. S15B). Confocal images were taken at different locations inside the construct to investigate whether the nanobeads could be perfused through the LVD-HUVEC networks.

Fig. S16 shows the representative images of nanobeads distribution in the inlet region and the inner regions. It indicates that only a few nanobeads appeared in the inlet region of the LVD tissue/gelatin gel construct (Fig. S16B). In contrast, no nanobeads were observed in the inner regions, demonstrating the occlusion of the tubular structures of LVD-HUVEC tissue. This is in accordance with the structural assessment in Fig. 2. It can be found that the interconnected tubular structures of confluent cells in LVD-HUVEC tissue were entirely filled with the heavily-compressed fibrin fibers, which induced the occlusion of the tubular structures.'

Minor Comments:

1. There are a few instances of writing tense being incorrect, please fix.

We have made corrections according to the reviewer's suggestions in the revised manuscript, which were highlighted with red in the main text.

REVIEWERS' COMMENTS

Reviewer #1 (Remarks to the Author):

The authors have addressed my concerns in the revised version of the manuscript.

I have four minor suggestions as further comment to the authors' responses.

We are sorry that the crucial reference was missed regarding the mechanical properties of the native myocardium tissues. The stress-strain curve of native heart tissues in Figure 6D is from the adult rat right ventricular myocardium, acquired from previous work (Nature Materials. 2008, 7(12):1003-10.). The relevant texts and reference have been updated in the revised manuscript, as below.

1) Please add in the figure legend (Fig 6) that the native tissue in panel D is from adult right ventricular myocardium.

We thank the reviewer for this comment. We chose the genes based on previous studies: 1) The genes, including GJA1, KCNJ2, PDK4, CPT1B, and PPARGC1A, have been successfully used for qRT-PCR to characterize the maturation of neonatal rat CM in the following studies: Nature communications, 2011, 2(1): 300; Nature biotechnology, 2013, 31(1): 54-62; Environmental health perspectives, 2012, 120(9): 1243-1251; Current Research in Physiology, 2022, 5: 55-62. 2) The genes, including S100A1, PLN, SCN5A, COX6A2, CKM, and PDK4, have been successfully used for qRT-PCR to characterize the maturation of hiPSC-CM in the following studies: Nature communications, 2017, 8(1): 1825; Frontiers in Cell and Developmental Biology, 2022, 10; Physiological genomics, 2012, 44(4): 245-258.)

2) Please cite these references in the manuscript.

Rat CMs or hiPSC-CMs

The LVD-cardiac tissues shown in Figure 6E-G are from neonatal rat cardiomyocytes. The relevant texts have been updated in the Result and Figure legend, ..

3) Please add this information to the Figure Legend (it was not updated in the revised manuscript) where it says cardiac, specify that it is from rat. This clarification is important because some constructs were made with hiPSC-CM, so it is important to make it clear in the figure legends which type of cells are being used in each instance.

We thank the reviewer for pointing it out. The stress-strain curve of native heart tissues is from adult rat right ventricular myocardium, acquired from previous work (Nature Materials. 2008, 7(12):1003-10.). The reference has been added in the revised manuscript, as below.

On Page 15, lines 339-340:

'...experimental measurement, and native adult rat heart tissue¹⁰, indicating that...'

4) Please add that it was acquired from the literature, this makes it clear that it is not experimental tissue.

Reviewer #2 (Remarks to the Author):

The authors clearly addressed concerns from both myself and the other reviewer. The presented work is markedly improved upon inspection. There is still a question as to the long term survivability of the tissue, along with mechanical integration into native tissue, but that would need to be addressed in a larger and separate animal study.

Response to Reviewers

We thank the reviewers for their constructive comments. The reviewers' comments are included in *red italics*, and our responses follow in regular font. In the main text, the changes are indicated in *red*.

Response to Reviewer #1:

1. The authors have addressed my concerns in the revised version of the manuscript.

I have four minor suggestions as further comment to the authors' responses.

We are sorry that the crucial reference was missed regarding the mechanical properties of the native myocardium tissues. The stress-strain curve of native heart tissues in Figure 6D is from the adult rat right ventricular myocardium, acquired from previous work (Nature Materials. 2008, 7(12):1003-10.). The relevant texts and reference have been updated in the revised manuscript, as below.

1) Please add in the figure legend (Fig 6) that the native tissue in panel D is from adult right ventricular myocardium.

We have made corrections according to the reviewer's suggestion in the revised manuscript. The relevant texts have been updated as below.

On Page 52, line 1079:

'...among the corresponding FEA calculations, experimental measurement, and adult rat right ventricular myocardium.'

2. We thank the reviewer for this comment. We chose the genes based on previous studies: 1) The genes, including GJA1, KCNJ2, PDK4, CPT1B, and PPARGC1A, have been successfully used for qRT-PCR to characterize the maturation of neonatal rat CM in the following studies: Nature communications, 2011, 2(1): 300; Nature biotechnology, 2013, 31(1): 54-62; Environmental health perspectives, 2012, 120(9): 1243-1251; Current Research in Physiology, 2022, 5: 55-62. 2) The genes, including S100A1, PLN, SCN5A, COX6A2, CKM, and PDK4, have been successfully used for qRT-PCR to characterize the maturation of hiPSC-CM in the following studies: Nature communications, 2017, 8(1): 1825; Frontiers in Cell and Developmental Biology, 2022, 10; Physiological genomics, 2012, 44(4): 245-258.)

2) Please cite these references in the manuscript.

We have cited these references in the revised manuscript.

3. Rat CMs or hiPSC-CMs

The LVD-cardiac tissues shown in Figure 6E-G are from neonatal rat cardiomyocytes. The relevant texts have been updated in the Result and Figure legend,..

3) Please add this information to the Figure Legend (it was not updated in the revised manuscript) where it says cardiac, specify that it is from rat. This clarification is important because some constructs were made with hiPSC-CM, so it is important to make it clear in the figure legends which type of cells are being used in each instance.

We have made corrections according to the reviewer's suggestion in the revised manuscript. The relevant texts have been updated as below.

On Page 52, lines 1080-1081:

'e Bright-field of the transferred monolayer LVD-rat-cardiac tissue. f,g The macroscopic (f) and confocal (g) images of the assembled 4-layer LVD-rat-cardiac construct...'

4. We thank the reviewer for pointing it out. The stress-strain curve of native heart tissues is from adult rat right ventricular myocardium, acquired from previous work (Nature Materials. 2008, 7(12):1003-10.). The reference has been added in the revised manuscript, as below.

On Page 15, lines 339-340:

'...experimental measurement, and native adult rat heart tissue¹⁰, indicating that...'

4) Please add that it was acquired from the literature, this makes it clear that it is not experimental tissue.

We have made corrections according to the reviewer's suggestion in the revised manuscript. The relevant texts have been updated as below.

On Page 15, line 342:

'...and native adult rat heart tissue (the data was acquired from the previous literature¹⁰)...'

Response to Reviewer #2:

The authors clearly addressed concerns from both myself and the other reviewer. The presented work is markedly improved upon inspection. There is still a question as to the long term survivability of the tissue, along with mechanical integration into native tissue, but that would need to be addressed in a larger and separate animal study.

We agree with the reviewer that a larger and separate animal study needs to be conducted in the future to investigate the long-term survivability of the LVD-cardiac tissues along with their mechanical integration into the host tissues post-implant. We expect to explore these crucial issues by implanting the LVD-hiPSC-derived-cardiac constructs from hiPSC-CMs onto the rat and porcine hearts for a long-term study. We have discussed these topics in the revised manuscript. The relevant texts have been updated in the Discussion section, as below.

On Page 20, lines 466-482:

‘Considering that previous work has demonstrated the ability of the tubular vasculature to template the neovascular architecture in vivo, we assume that the LVD-HUVEC tissue can guide the formation of neo-vasculature with the biomimetic hierarchical architecture and induce ingrowth and anastomoses of host vasculature in vivo which can facilitate the mass transfer. Therefore, a large and separate animal study needs to be conducted in the future to investigate the growth of LVD-HUVEC tissues and their effect on the long-term survivability of the LVD-cardiac tissues post-implant.

The mechanical integration of the LVD-cardiac tissues into the host tissues also needs further investigation through animal study. Previous works have demonstrated the benefits of mechanical support from biomaterial-based cardiac patches, which enable to reduce myocardium wall stress and subsequently limit adverse remodeling and improve cardiac function after myocardial infarction⁶⁰. However, the optimum patch design is still unknown. In our study, EHD printing was employed to produce the elastic PCL scaffold with precise microscale serpentine structures, thus granting us freedom in tailoring the mechanical property of the LVD-cardiac tissue constructs. This allows us to study the effect of cardiac patches with programmed anisotropic mechanical properties on the heart’s remodeling process and pumping function. For this purpose, the LVD-cardiac tissue constructs should be implanted onto the hearts of large animal models, such as pigs and dogs.’

On Page 23, lines 527-529:

‘In the future, we expect to engineer the functional LVD-hiPSC-derived-cardiac constructs *in vitro* and conduct large-scale and long-term animal studies to investigate their post-implant survivability and mechanical integration into the host tissues.’